# The protective roles of eugenol on type 1 diabetes mellitus through NRF2-mediated oxidative stress pathway

Yalan Jiang[1†], Pingping He[1†], Ke Sheng[2†], Yongmiao Peng[2], Huilan Wu[2], Songwei Qian[3,4], Weiping Ji[3,4*‡], Xiaoling Guo[1,2,5,6*‡], Xiaoou Shan[1,5,6*‡]

[1]Department of Pediatrics, the Second Affiliated Hospital and Yuying Children's Hospital of Wenzhou Medical University, Wenzhou, China; [2]Basic Medical Research Center, the Second Affiliated Hospital and Yuying Children's Hospital of Wenzhou Medical University, Wenzhou, China; [3]Department of Genaral Surgery, the Quzhou Affiliated Hospital of Wenzhou Medical University, Quzhou People's Hospital, Quzhou, China; [4]Department of General Surgery, the Second Affiliated Hospital and Yuying Children's Hospital of Wenzhou Medical University, Wenzhou, China; [5]Key Laboratory of Children Genitourinary Diseases of Wenzhou, the Second Affiliated Hospital and Yuying Children's Hospital of Wenzhou Medical University, Wenzhou, China; [6]Key Laboratory of Structural Malformations in Children of Zhejiang Province, the Second Affiliated Hospital and Yuying Children's Hospital of Wenzhou Medical University, Wenzhou, China

*For correspondence:
jiweiping@wmu.edu.cn (WJ);
guoxling@hotmail.com (XG);
Seagullshan@wmu.edu.cn (XS)

†These authors contributed equally to this work
‡These authors also contributed equally to this work

## eLife Assessment

This **useful** study partially succeeds in providing **solid** evidence in support of the therapeutic potential of the plant-derived compound eugenol for ameliorating symptoms associated with STZ-induced oxidative stress, identifying Nuclear factor E2-related factor (Nrf2) as a mediator of the effects induced by eugenol. Although the study provides interesting data, there remain concerns associated with the STZ model and the rather superficial mechanistic assessment.

**Abstract** Type 1 diabetes mellitus (T1DM), known as insulin-dependent diabetes mellitus, is characterized by persistent hyperglycemia resulting from damage to the pancreatic β cells and an absolute deficiency of insulin, leading to multi-organ involvement and a poor prognosis. The progression of T1DM is significantly influenced by oxidative stress and apoptosis. The natural compound eugenol (EUG) possesses anti-inflammatory, anti-oxidant, and anti-apoptotic properties. However, the potential effects of EUG on T1DM had not been investigated. In this study, we established the streptozotocin (STZ)-induced T1DM mouse model in vivo and STZ-induced pancreatic β cell MIN6 cell model in vitro to investigate the protective effects of EUG on T1DM, and tried to elucidate its potential mechanism. Our findings demonstrated that the intervention of EUG could effectively induce the activation of nuclear factor E2-related factor 2 (NRF2), leading to an up-regulation in the expressions of downstream proteins NQO1 and HMOX1, which are regulated by NRF2. Moreover, this intervention exhibited a significant amelioration in pancreatic β cell damage associated with T1DM, accompanied by an elevation in insulin secretion and a reduction in the expression levels of apoptosis and oxidative stress-related markers. Furthermore, ML385, an NRF2 inhibitor, reversed these effects of EUG. The present study suggested that EUG exerted protective effects on pancreatic β cells in T1DM by attenuating apoptosis and oxidative stress through the activation of the NRF2 signaling pathway. Consequently, EUG holds great promise as a potential therapeutic candidate for T1DM.

## Introduction

Type 1 diabetes mellitus (T1DM) is a chronic and progressive autoimmune disorder characterized by severe destruction of pancreatic β cells, leading to an absolute deficiency of insulin and subsequent hyperglycemia (*Lu and Zhao, 2020*). The symptoms of this disease include polydipsia, polyphagia, polyuria, wasting, etc., which can progress to ketoacidosis and even result in death without any treatment (*Syed, 2022*). Additionally, chronic hyperglycemia can cause various complications, including diabetic nephropathy (*Papadopoulou-Marketou et al., 2018*) and diabetic cardiomyopathy (*Wang et al., 2022a*), which impose huge social and economic burdens. The global incidence of T1DM is rapidly increasing. According to the International Diabetes Federation (IDF), it is estimated that there were approximately 537 million adults (aged 20–79) worldwide affected by diabetes in 2021, while over 1.2 million children and adolescents (aged 0–19) were diagnosed with T1DM. The global prevalence of diabetes is projected to reach approximately 783 million individuals by 2045, with developing countries experiencing the largest increase (*Magliano and Boyko, 2021*). Currently, insulin intervention remains the most efficacious approach for managing T1DM. However, challenges persist in addressing postprandial glycemic fluctuations (*Cobry et al., 2010*) and life expectancy is shorter. Furthermore, in comparison to the general population, the quality of life is considerably compromised (*Kalyva et al., 2011*; *Cameron et al., 2002*; *Miller et al., 2012*). Therefore, it is crucial to explore more effective treatment strategies for T1DM patients.

Oxidative stress, a widely discussed concept in recent years, refers to the imbalance between oxygen free radical production and anti-oxidant reactions within cells, playing a crucial role in diabetes pathogenesis (*Rains and Jain, 2011*). Hyperglycemia in diabetic patients can result in an augmented generation of reactive oxygen species (ROS) and the up-regulation of oxidative stress-related markers in vivo, while decreasing expression of anti-oxidant markers and disrupting internal homeostasis (*Maiese et al., 2007*). The cells typically possess robust mechanisms for anti-oxidant protection to maintain cellular homeostasis, particularly through the action of nuclear factor-E2-related factor 2 (NRF2), which serves as an efficient regulator of cellular anti-oxidant defense by modulating the expression of genes encoding anti-oxidant proteins (*Sajadimajd and Khazaei, 2018*). NRF2 can interact with the Kelch-like ECH-associated protein 1 (KEAP1) in the cytoplasm under normal physiological conditions, leading to subsequent degradation through ubiquitination. When cells are exposed to oxidative stress, NRF2 dissociates from KEAP1, evades ubiquitination, and translocates from the cytoplasm into the nucleus. Subsequently, it enhances the expression of target proteins, such as NAD(P)H quinone oxidoreductase-1 (NQO1) and heme oxygenase-1 (HMOX1) by binding to the anti-oxidant response element (ARE) (*Dinkova-Kostova et al., 2002*; *Nguyen et al., 2009*; *Wasserman and Fahl, 1997*; *Zhang and Hannink, 2003*; *Zhu et al., 2005*). Furthermore, ROS can induce DNA damage, and modulate the expression of proteins involved in cell regulation inhibition or activation, impair mitochondrial function, ultimately leading to apoptosis (*Aguiar et al., 2013*; *Liu et al., 2021*). Recent findings have indicated that the systemic activation of NRF2 signaling leads to a delay in the onset of T1DM in spontaneous non-obese diabetic mice models (*Yagishita et al., 2019*). Therefore, targeting NRF2 may offer potential for the prevention and treatment of T1DM.

Eugenol (EUG), a phenolic aromatic compound, is primarily derived from clove oil. The compound exhibits a range of biological activities, including anti-oxidant, anti-inflammatory, anti-apoptotic, and anti-bacterial effects (*Ulanowska and Olas, 2021*). Previous study has demonstrated that EUG could alleviate colitis by reducing oxidative stress through the activation of the KEAP1-NRF2 signaling pathway (*Chen et al., 2021*). Additionally, EUG has been observed to attenuate apoptosis induced by transmissible gastroenteritis virus through the ROS-NRF2-ARE signaling pathway (*Wang et al., 2022b*).

So far, no reports have been published to ascertain whether EUG confers a protective effect on damaged pancreatic β cells in individuals with T1DM. Our study aims to investigate the protective effect of EUG on pancreatic β cell damage in streptozotocin (STZ)-induced T1DM mouse model and STZ-induced MIN6 cell model, with the objective of elucidating the underlying mechanism. The findings may provide a potential novel therapeutic candidate drug for T1DM.

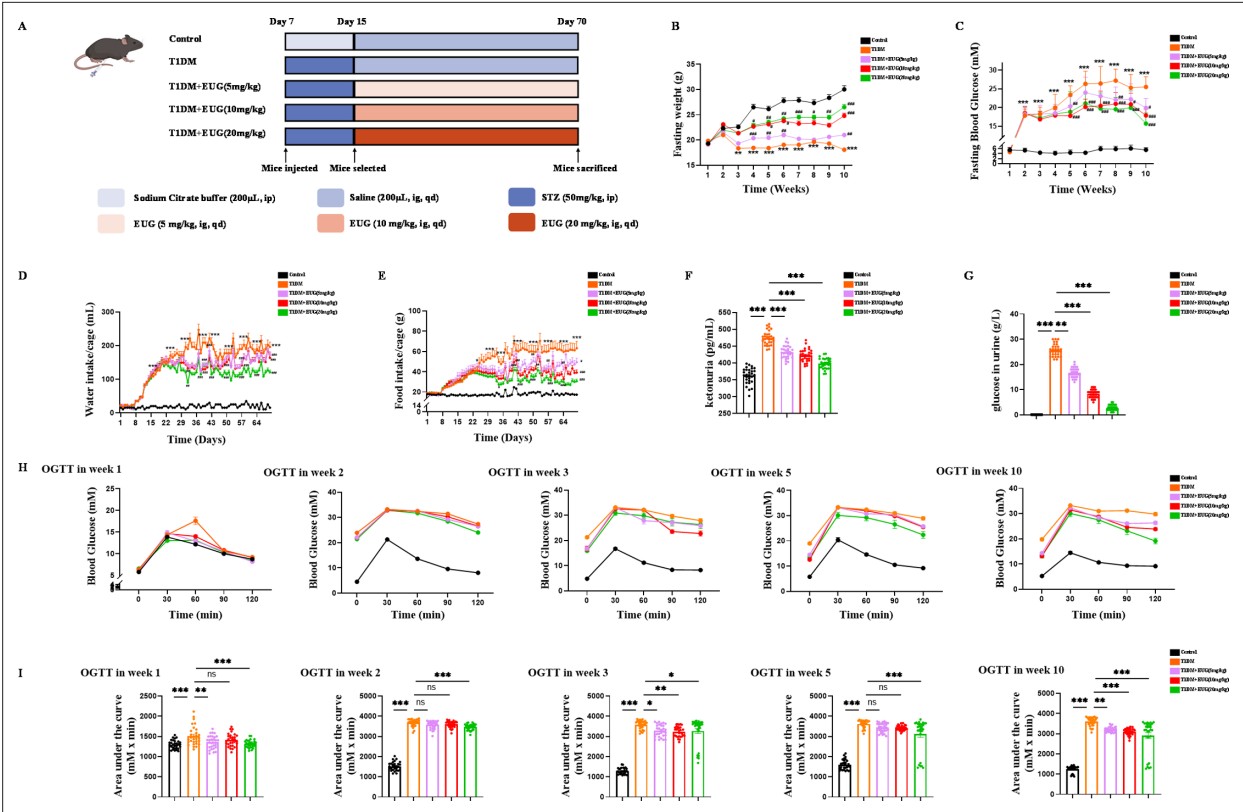

**Figure 1.** The eugenol (EUG) treatment effectively alleviated symptoms associated with type 1 diabetes mellitus (T1DM) mice. (**A**) The schematic diagram depicts the progress of animal experiments in each group of mice. Different colors indicate different treatments for mice. (**B**) The fasting weight levels of mice were measured weekly in each group (n=30 mice). (**C**) The fasting blood glucose levels of mice were measured weekly in each group (n=30 mice). (**D**) The water intake/cage in each group of mice (n=6 cages, 5 mice/cage). (**E**) The food intake/cage in each group of mice (n=6 cages, 5 mice/cage). (**F**) Urine ketones in each group were detected by enzyme-linked immunosorbent assay (ELISA) (n=30 mice). (**G**) The urine glucose levels of mice were measured by biochemical test in each group of mice (n=30 mice). (**H**) The curve graph of oral glucose tolerance test (OGTT) from 0 min to 120 min at week 1, week 2, week 3, week 5, and week 10 (n=30 mice). (**I**) The quantitative results of OGTT at week 1, week 2, week 3, week 5, and week 10 (n=30 mice). Mean ± SEM. All experiments were repeated at least three times independently. Compare with the Control group *p<0.05, compare with the Control group **p<0.01, compare with the Control group ***p<0.001, compare with the T1DM group #p<0.05, compare with the T1DM group ##p<0.01, compare with the T1DM group ###p<0.001 indicate significant differences, and ns>0.05 means no significance difference. One-way ANOVA.

The online version of this article includes the following figure supplement(s) for figure 1:

**Figure supplement 1.** Eugenol (EUG) alleviated the related complications in type 1 diabetes mellitus (T1DM) mice.

## Results

### EUG can relieve the symptoms associated with T1DM and reduce the blood glucose level in T1DM mice

The experimental design timeline for in vivo experiments was depicted in *Figure 1A*. To assess the effects of EUG on T1DM mice, we recorded the fasting body weight (*Figure 1B*), fasting blood glucose levels (*Figure 1C*), water intake in 24 hr (*Figure 1D*), food intake in 24 hr (*Figure 1E*), and area of urine-soaked pads in T1DM mice (*Figure 1—figure supplement 1A*). These results showed that EUG effectively alleviated the multiple symptoms associated with T1DM, including polydipsia, hyperphagia, polyuria, and weight loss, while 20 mg/kg EUG exhibited the better improvement. At the termination of the experiment, the mice in each group showed distinct growth states (*Figure 1—figure supplement 1B*). The results demonstrated that the mice in T1DM group were smaller than the mice in Control group, and EUG treatment could improve this phenomenon. Given that T1DM can elevate the levels of urine glucose and urine ketone, we conducted biochemical analysis for urine ketone and urine glucose (*Figure 1—figure supplement 1C*) in each group of mice. The results

(*Figure 1F and G*) showed that administration of EUG exhibited a mitigating effect on the elevation of urine ketone and urine glucose levels induced by T1DM.

It is widely acknowledged that oral glucose tolerance test (OGTT) can be used to detect T1DM and predict prognosis (*Helminen et al., 2015*). In this study, mice in each group were performed OGTT at week 1 (prior to T1DM induction), week 2 (2 days after T1DM induction), week 3 (1 week after T1DM induction), week 5 (3 weeks after T1DM induction), and week 10 (at the end of the experiment). The findings demonstrated that treatment with EUG resulted in a reduction in blood glucose levels and improvement in islet function in T1DM mice (*Figure 1H and I*). The chronic and progressive nature of diabetes necessitates vigilance, as prolonged hyperglycemia can exert deleterious effects on vital organs such as the heart, kidneys, and liver (*Papadopoulou-Marketou et al., 2018*; *Wang et al., 2022a*; *Khoury et al., 2018*). Therefore, periodic acid-schiff (PAS) staining was performed on the kidneys (*Figure 1—figure supplement 1D*). The results showed that T1DM indeed induced glycogen accumulation within the glomerulus, while EUG intervention showed reduction in such glycogen accumulation, thereby enhancing the prognosis of T1DM. In brief, the administration of EUG could relieve the symptoms and reduce the blood glucose level in T1DM mice.

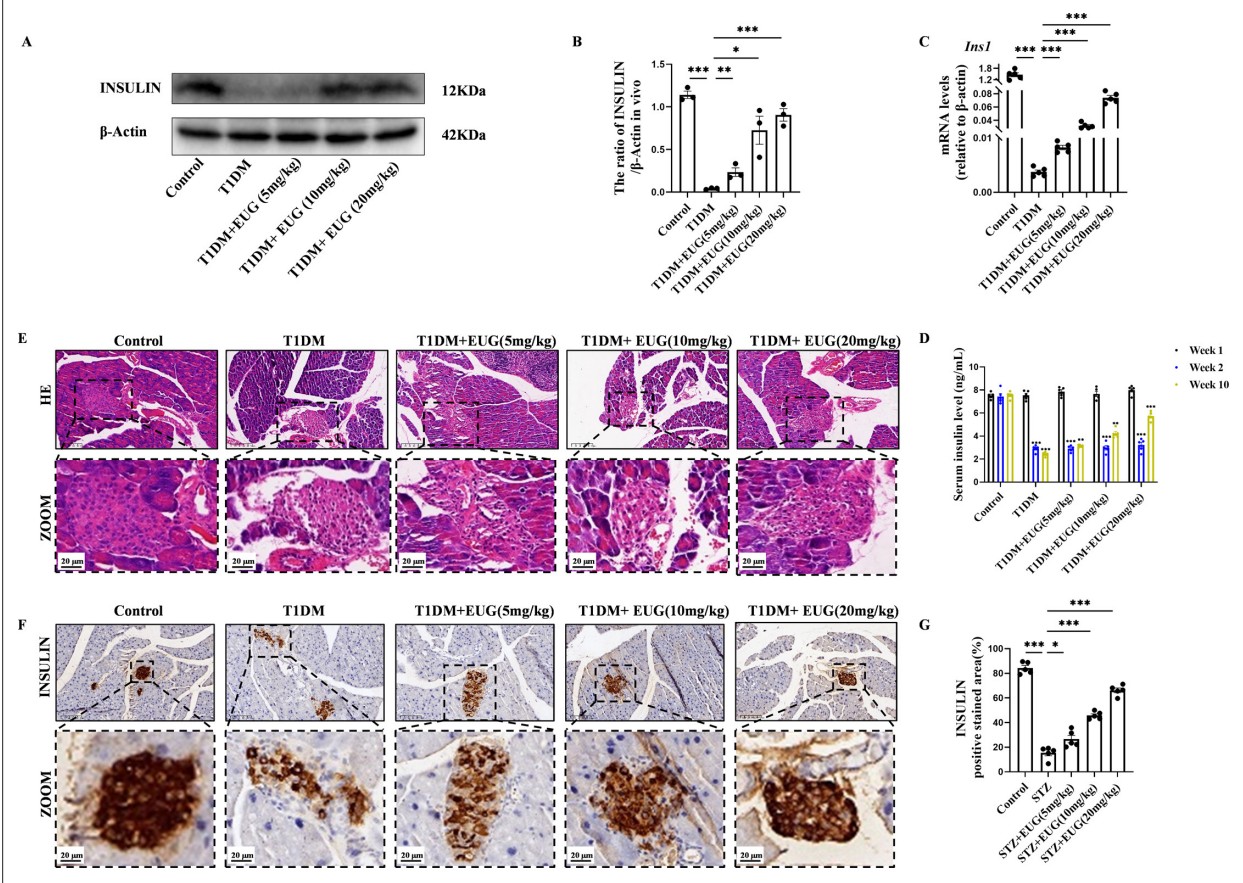

**Figure 2.** Eugenol (EUG) improved the pancreas islet structure and function in type 1 diabetes mellitus (T1DM) mice. (**A**) The detection of insulin expression in different groups using western blot. (**B**) The quantification of western blot gel bands in different groups (n=3 independently repeated experiments). (**C**) The gene levels of *Ins1* in different groups (n=5 independently repeated experiments). (**D**) Enzyme-linked immunosorbent assay (ELISA) analysis of serum fasting insulin levels at different time points (n=5 independently repeated experiments). (**E**) The representative hematoxylin and eosin (H&E) staining images of pancreatic paraffin sections in each group of mice. Scale bars 50 µm, 20 µm. (**F**) The representative immunohistochemical staining of insulin in pancreas islet in each group of mice. Scale bars 50 µm, 20 µm. (**G**) The quantitative analysis of immunohistochemical staining (n=5 independently repeated experiments). Mean ± SEM. *p<0.05, **p<0.01, ***p<0.001 indicate significant differences, and ns>0.05 means no significance difference. One-way ANOVA.

The online version of this article includes the following source data for figure 2:

**Source data 1.** Raw unedited gels for *Figure 2*.

**Source data 2.** Uncropped and labeled gels for *Figure 2*.

## EUG can improve the damage degree of islets in T1DM mice

Insulin is produced by the β cells of pancreatic islets. Western blot (*Figure 2A*) and RT-qPCR were evaluated and used to assess insulin expression levels in each group of mice. The quantitative results from both western blot (*Figure 2B*) and RT-qPCR (*Figure 2C*) showed that there was a decrease in *Ins1* expression levels in T1DM mice, whereas EUG intervention increased the expression of insulin. In addition, the serum insulin levels of mice in each group were quantified using enzyme-linked immunosorbent assay (ELISA) at different time points. The results showed a reduction in serum insulin levels after T1DM modeling, which were subsequently recovered by EUG intervention. Notably, the administration of 20 mg/kg EUG exhibited the better improvement effect (*Figure 2D*). The islets' structural integrity in each group of mice was performed through hematoxylin and eosin (H&E) staining. The results revealed that, in comparison to the Control group, the islet structure of T1DM mice exhibited severe damage with indistinct boundaries and evident vacuolization in the islet cells. EUG intervention demonstrated a potential for ameliorating the extent of islet damage (*Figure 2E*). Furthermore, insulin immunohistochemical staining was performed to further evaluate the expression of insulin in each group of mice (*Figure 2F*). The quantitative results showed a enhancement in insulin expression among T1DM mice following EUG treatment, and 20 mg/kg EUG displayed a better improvement (*Figure 2G*). These findings indicated that intervention with EUG could effectively ameliorate islet damage in T1DM mice.

## EUG intervention alleviates T1DM by improving oxidative stress pathways

To investigate the potential molecular mechanisms underlying the regulation in T1DM, we performed RNA-seq analysis on pancreatic tissues obtained from the Control group, T1DM group, and EUG intervention group. The RNA-seq data revealed that among differentially expressed genes (DEGs), 28 genes were up-regulated, and 17 genes were down-regulated in the EUG group compared to the T1DM group (*Figure 3—figure supplement 1A*). Furthermore, a heat map was generated to visualize the relative abundance of these DEGs affected by T1DM (*Figure 3—figure supplement 1B*). The findings were consistent with previous results, which showed the intervention of EUG could enhance the functionality of islet β cells, augment insulin secretion, and mitigate hyperglycemia in T1DM mice. Gene set enrichment analysis (GSEA) based on BP (biological process) gene revealed that compared to the T1DM group, the EUG intervention group exhibited significant enrichment in pathways, including 'response to glucose', 'response to carbohydrate', 'cellular glucose homeostasis', and 'positive regulation of insulin secretion' with NES>0 (*Figure 3—figure supplement 1C–F*). These findings confirmed our previous experimental results and suggested a beneficial effect of EUG intervention on T1DM. To explore the therapeutic mechanism of EUG further, GSEA based on MF (molecular function) gene set was performed. The results demonstrated that compared to the T1DM group, the EUG group showed significant enrichment and negative regulation in pathways related to 'hydrogen peroxide-mediated programmed cell death' and 'cellular response to hydrogen peroxide' with NES<0 (*Figure 3—figure supplement 1G and H*). Therefore, we concluded that EUG may alleviate T1DM by improving oxidative stress pathway.

## EUG protects pancreatic β cells in T1DM mice by activating the NRF2 signaling pathway

Oxidative stress as a prominent pathogenic mechanism in T1DM is widely recognized (*Piganelli et al., 2020*). In addition, the EUG compound has been reported to possess potent anti-oxidant properties and effectively activate the nuclear factor E2-related factor 2 (NRF2) (*Ma et al., 2021*). Western blot was used to detect the expression levels of total NRF2 protein (T-NRF2) and nuclear NRF2 protein (N-NRF2) in each group of mice (*Figure 3A*). The findings showed that, in comparison to the Control group, intervention with EUG could activate NRF2, exerting anti-oxidative effects (*Figure 3B*). The result of RT-qPCR was consistent with those of western blot (*Figure 3C*). Furthermore, western blot was conducted to evaluate the expression of key proteins involved in the NRF2 signaling pathway in each group of mice, including Kelch-like ECH-associated protein 1 (KEAP1), heme oxygenase-1 (HMOX1), and NAD(P)H quinone dehydrogenase 1(NQO1) (*Figure 3D*). The results showed that the intervention of EUG led to an up-regulation in protein expression levels of HMOX1 and NQO1, while a down-regulation in KEAP1 expression was observed due to the activation of the NRF2 signaling

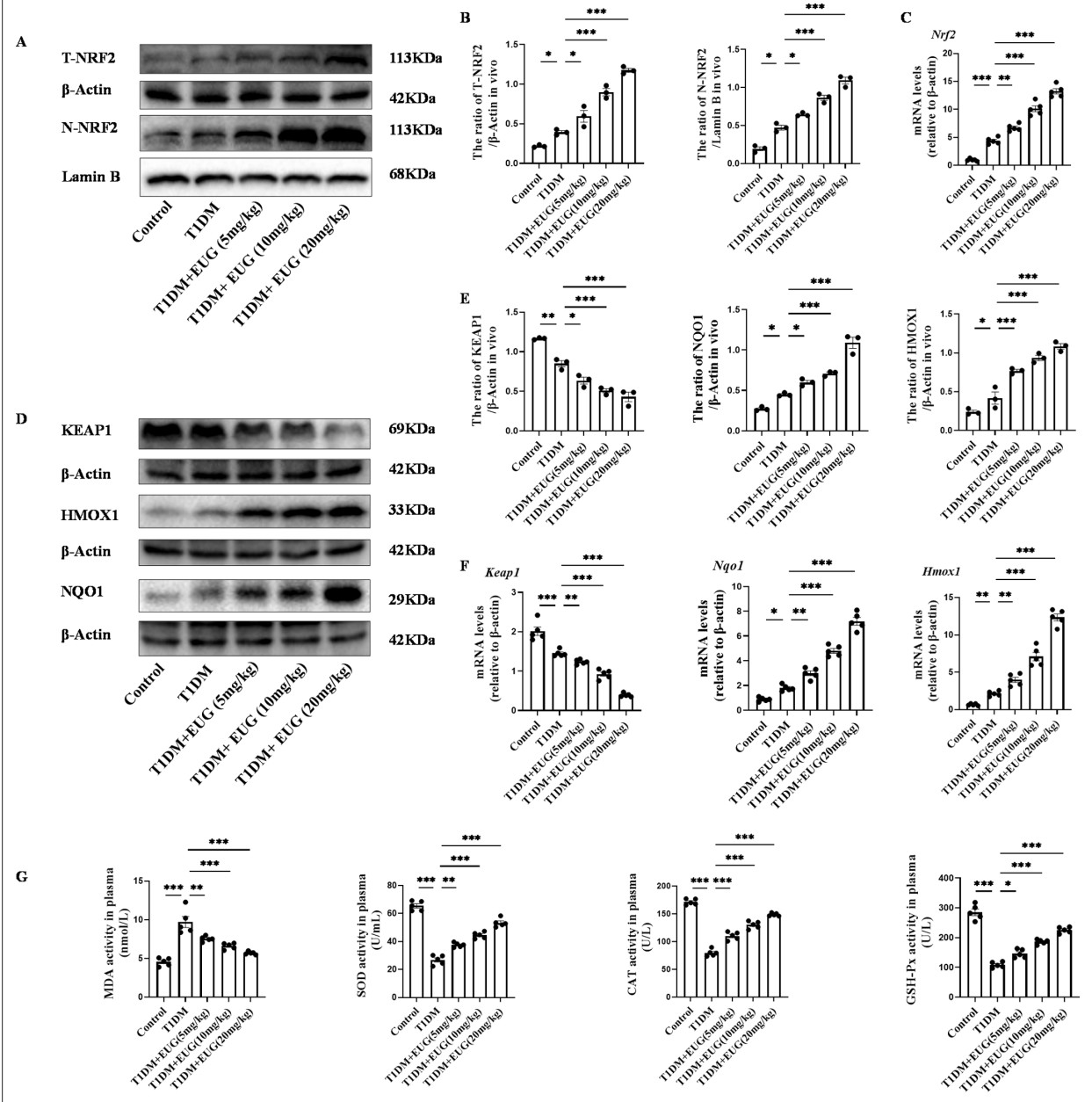

**Figure 3.** Eugenol (EUG) attenuated excessive oxidative stress through activating nuclear factor E2-related factor 2 (NRF2) signaling pathway in type 1 diabetes mellitus (T1DM) mice. (**A**) The detection of T-NRF2, N-NRF2 expression in different groups using western blot. (**B**) The quantification of western blot gel bands in different groups (n=3 independently repeated experiments). (**C**) The gene levels of *Nrf2* in different groups (n=5 independently repeated experiments). (**D**) The detection of KEAP1, HMOX1, and NQO1 expression in different groups using western blot. (**E**) The quantification of western blot gel bands in different groups (n=3 independently repeated experiments). (**F**) The gene levels of *Keap1, Nqo1,* and *Hmox1* in different groups (n=5 independently repeated experiments). (**G**) The levels of serum biochemical indexes (malondialdehyde [MDA], superoxide dismutase [SOD], catalase [CAT], and glutathione peroxidase [GSH-Px]) in each group of mice (n=5 independently repeated experiments). Mean ± SEM. *p<0.05, **p<0.01, ***p<0.001 indicate significant differences, and ns>0.05 means no significance difference. One-way ANOVA.

The online version of this article includes the following source data and figure supplement(s) for figure 3:

**Source data 1.** Raw unedited gels for *Figure 3*.

**Source data 2.** Uncropped and labeled gels for *Figure 3*.

**Figure supplement 1.** Statistical map of mRNA differential expression between type 1 diabetes mellitus (T1DM) group and eugenol (EUG) intervention group.

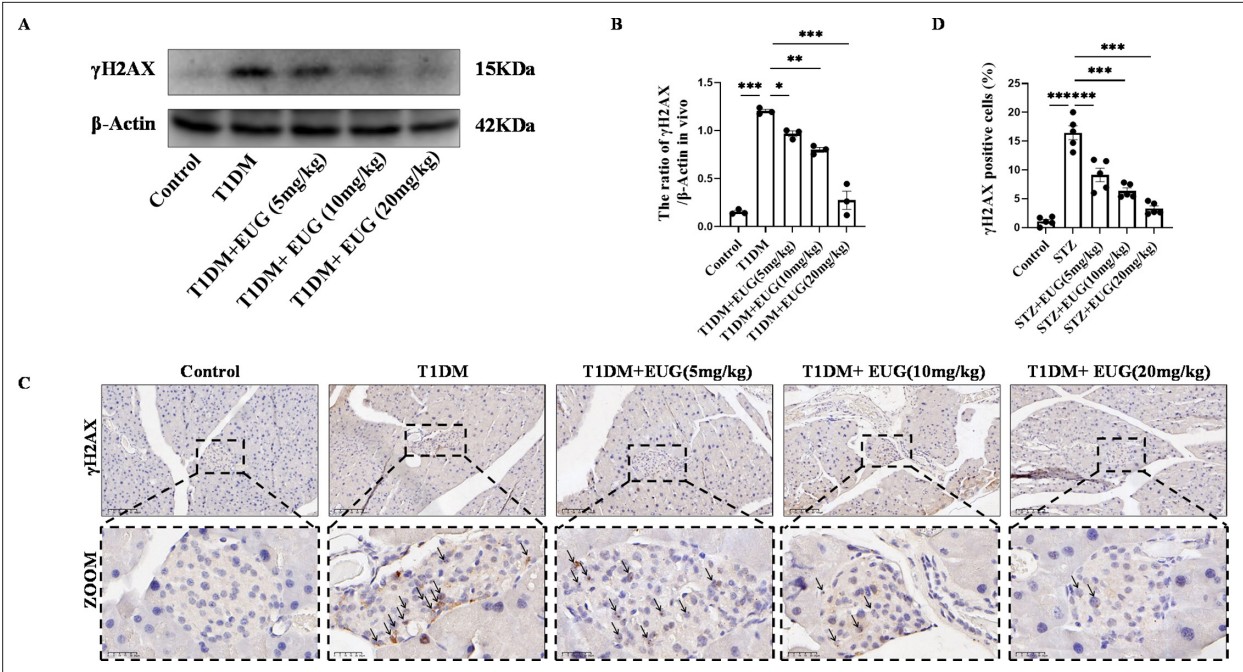

**Figure 4.** Eugenol (EUG) decreased the expression level of γH2AX in type 1 diabetes mellitus (T1DM) mice. (**A**) The detection of γH2AX expression in different groups using western blot. (**B**) The quantification of western blot gel bands in different groups (n=3 independently repeated experiments). (**C**) The representative immunohistochemical staining of γH2AX in pancreas islet in each group of mice. Black arrows were employed to highlight the presence of brown-stained islet β cells. Scale bars 100 μm, 25 μm. (**D**) The quantitative analysis of immunohistochemical staining (n=5 independently repeated experiments). Mean ± SEM. *p<0.05, **p<0.01, ***p<0.001 indicate significant differences, and ns>0.05 means no significance difference. One-way ANOVA.

The online version of this article includes the following source data for figure 4:

**Source data 1.** Raw unedited gels for *Figure 4*.

**Source data 2.** Uncropped and labeled gels for *Figure 4*.

pathway (*Figure 3E*). The results of *Keap1, Nqo1,* and *Hmox1* gene expression were consistent with aforementioned protein expression (*Figure 3F*). Finally, biochemical assays were conducted to measure the activities of serum oxidative stress-related markers such as malondialdehyde (MDA), superoxide dismutase (SOD), catalase (CAT), and glutathione peroxidase (GSH-Px) in each group of mice. The oxidative stress-related index (MDA) was found to be elevated in the T1DM group, while the expression of anti-oxidant stress-related indexes (SOD, CAT, and GSH-Px) were increased in the EUG intervention group (*Figure 3G*). These data suggested that EUG has the potential to alleviate oxidative stress-induced damage to pancreatic β cells in TIDM mice through activating the NRF2 signaling pathway.

## EUG can reduce the apoptosis of pancreatic β cells in T1DM mice

DNA is essential for cell survival, and γH2AX serves as a dependable biomarker for detecting DNA damage (*Kinner et al., 2008*). Western blot was performed to assess the expression of γH2AX in each group of mice (*Figure 4A*). The result showed that the pancreatic β cells of T1DM mice exhibited DNA damage, and intervention with EUG was found to reduce the extent of DNA damages (*Figure 4B*). γH2AX immunohistochemical staining was also performed to assess γH2AX expression in each group of mice (*Figure 4C*). The quantitative finding was consistent with the western blot (*Figure 4D*).

It is well known that DNA damage in cells is closely associated with apoptosis. The potential anti-apoptotic effects of EUG on mouse pancreatic β cells were investigated by conducting a western blot to assess the expression levels of BCL2, BAX, and Cleaved Caspase-3 in the pancreatic β cells of mice in each experimental group (*Figure 5A*). The results demonstrated that there was a increase in the number of apoptotic pancreatic β cells in T1DM mice compared to Control group. Moreover, EUG intervention effectively suppressed the apoptosis of pancreatic β cells in T1DM mice as well as

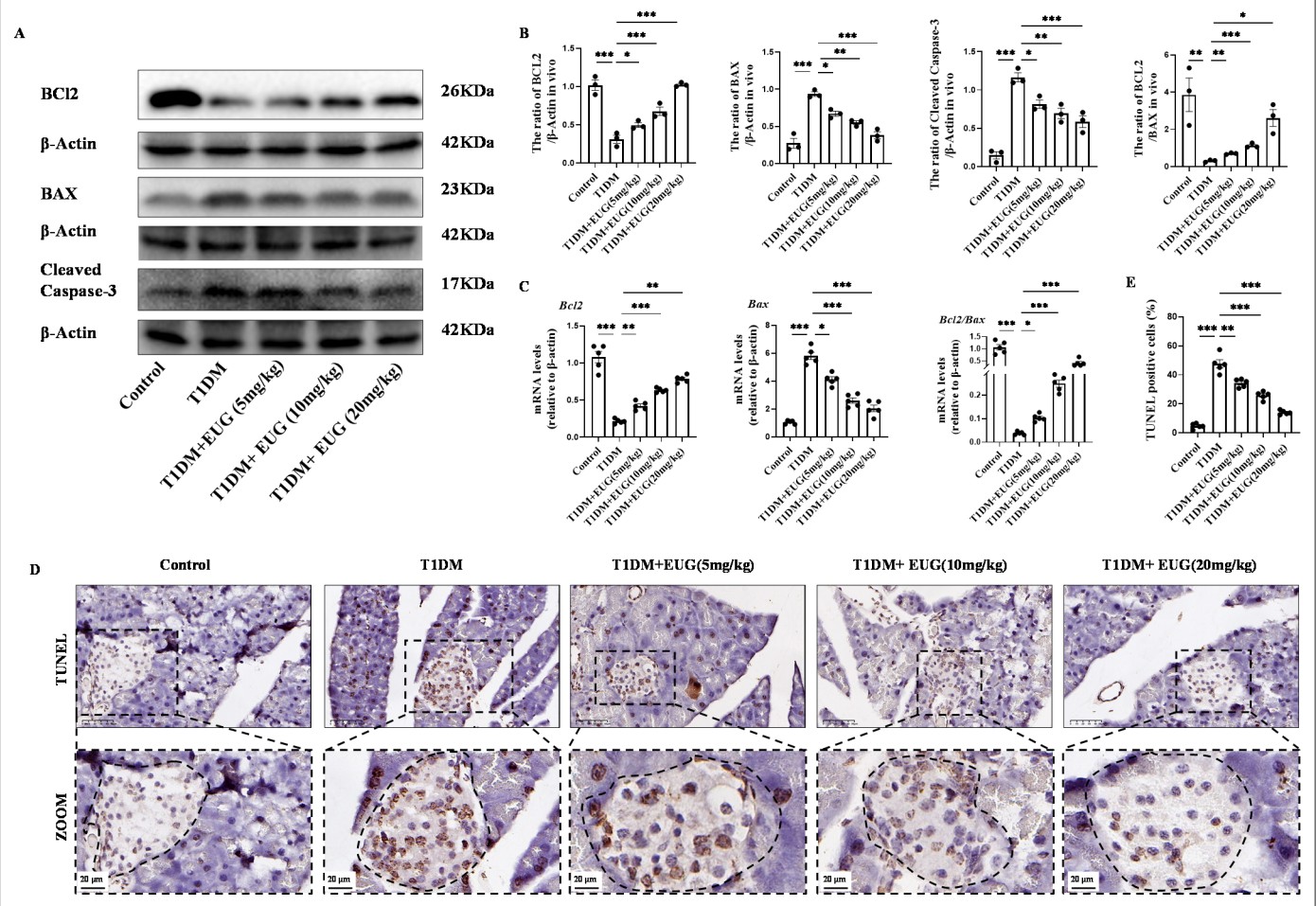

**Figure 5.** Eugenol (EUG) reduced apoptosis of pancreatic β cells in type 1 diabetes mellitus (T1DM) mice. (**A**) The detection of BCL2, BAX, Cleaved Caspase-3 expression in different groups using western blot. (**B**) The quantification of western blot gel bands in different groups (n=3 independently repeated experiments). (**C**) The gene levels of *Bax*, *Bcl2*, and *Bcl2/Bax* in different groups (n=5 independently repeated experiments). (**D**) The immunohistochemical staining of TUNEL in pancreas islet in each group of mice. Scale bars 50 μm, 20 μm. (**E**) The quantification results of TUNEL staining in different groups (n=5 independently repeated experiments). Mean ± SEM. *p<0.05, **p<0.01, ***p<0.001 indicate significant differences, and ns>0.05 means no significance difference. One-way ANOVA.

The online version of this article includes the following source data for figure 5:

**Source data 1.** Raw unedited gels for *Figure 5*.

**Source data 2.** Uncropped and labeled gels for *Figure 5*.

20 mg/kg EUG had the better effects (*Figure 5B*). Furthermore, the expression levels of *Bcl2* and *Bax* genes in pancreatic β cells of mice in each group were detected by RT-qPCR. The results obtained from western blot were consistent with these findings (*Figure 5C*). Finally, TUNEL staining was further performed to evaluate the apoptosis of pancreatic β cells in each group of mice (*Figure 5D*). The quantitative result showed an elevated count of apoptotic pancreatic β cells in T1DM mice, which could be mitigated by EUG intervention (*Figure 5E*). These findings suggested that EUG exhibited a potential to attenuate apoptosis of pancreatic β cells in T1DM mice.

## EUG can alleviate the impairment of STZ-induced MIN6 cells

To further explore the potential effects of EUG on T1DM, we established STZ-induced MIN6 cell model in vitro. The cytotoxicity of STZ on MIN6 cells was evaluated using the CCK-8 assay. The result showed a dose-dependent decrease in the viability of MIN6 cells after 24 hr of treatment with various concentrations of STZ (0–8 mM), and the optimal concentration for STZ treatment was 1 mM (*Figure 6—figure supplement 1A*). In addition, the cytotoxicity of EUG on MIN6 cells was

also assessed using the CCK-8 assay. The result showed that the cell viability was not significantly affected by EUG concentration within the range of 0–600 µM (*Figure 6—figure supplement 1B*). After being treated with various concentrations of EUG (0–400 µM) for 2 hr, MIN6 cells were subjected to 1 mM STZ treatment. The result showed that pre-treatment with EUG could enhance the viability of MIN6 cells, and reached the optimum after 50 µM EUG treatment (*Figure 6—figure supplement 1C*). Optical microscopy observation revealed that compared to Control group, the number of cells in STZ group was significantly reduced and cell morphology was worse. The administration of EUG treatment could ameliorate the above phenomena, whereas the intervention of NRF2 inhibitor ML385 (10 µM) could aggravate the cell damage (*Figure 6—figure supplement 1D*).

The experimental design timeline for in vitro experiments was illustrated in *Figure 6A*. Given the potential impact of pancreatic β cell damage on insulin secretion, we conducted an assessment of insulin protein and gene expression levels in different groups. The expression levels of insulin protein and gene in each group were assessed using western blot (*Figure 6B*) and RT-qPCR. The results (*Figure 6C and D*) showed that insulin levels in the STZ-induced group were lower than those in the Control group. However, after EUG treatment, STZ-induced insulin levels in MIN6 cells were elevated, which could be reversed by the intervention of the NRF2 inhibitor ML385. Furthermore, the cell culture supernatant of each group was further collected for detecting insulin levels using ELISA, and the obtained result was consistent with the aforementioned findings (*Figure 6E*). These findings suggested that EUG has the potential to enhance the insulin secretion of STZ-induced MIN6 cells.

## EUG reduces STZ-induced MIN6 cell damage through activating the NRF2 signaling pathway

In order to further investigate the potential mechanism of EUG on T1DM, ML385, the typical NRF2 antagonist, was employed to assess the expression of proteins related to the NRF2 pathway through western blot (*Figures 6F and 7A*). The quantitative results showed that the protein levels of T-NRF2/β-actin and N-NRF2/lamin B in MIN6 cells treated with EUG were higher than those in the Control group (*Figure 6H*). Furthermore, treatment with EUG also resulted in an elevation of HMOX1 and NQO1 protein levels, while simultaneously reducing the level of KEAP1 (*Figure 7B*). However, the administration of ML385 could effectively reverse the effects of EUG on STZ-induced MIN6 cells. The quantitative results demonstrated that the ratio of T-NRF2/β-actin and N-NRF2/lamin B decreased after ML385 treatment. Moreover, RT-qPCR (*Figures 6I and 7C*) and immunofluorescence staining (*Figures 6G and 7D*) were conducted to evaluate the expressions of HMOX1 and NRF2 in MIN6 cells under different treatments, with the trends of results were consistent with those of obtained from western blot and RT-qPCR (*Figures 6J and 7F*).

In order to demonstrate the ability of EUG to alleviate the STZ-induced MIN6 cells in vitro by activating the NRF2 signaling pathway, we utilized the NRF2 inhibitor ML385 to assess oxidative stress among different groups. MitoSOX staining (*Figure 7E*) showed that EUG reduced mitochondrial ROS levels in STZ-induced MIN6 cells, whereas ML385 was found to weaken this effect (*Figure 7G*). In addition, the flow cytometry was used to detect the cell ROS levels in different groups (*Figure 7H*). The quantitative result showed that STZ could increase the level of cell ROS, but EUG intervention could reduce the cell ROS level in STZ-induced MIN6 cells. Similarly, ML385 could reverse EUG effect on STZ-induced MIN6 (*Figure 7I*). These data suggested that EUG has the potential to ameliorate oxidative stress-induced damage in MIN6 cells caused by activating the NRF2 signaling pathway.

## EUG inhibited STZ-induced apoptosis of islet β cell MIN6

To further explore the potential of EUG in attenuating STZ-induced damage to MIN6 cells in vitro through activating of the NRF2 signaling pathway, we used NRF2 inhibitor ML385 to evaluate the apoptosis levels in MIN6 cells under different treatments. Western blot result of γH2AX (*Figure 8A*) revealed that STZ treatment induced an increase in DNA damage in MIN6 cells, while intervention with EUG effectively reduced the extent of DNA damage in MIN6 cells, and ML385 intervention could reversed this phenomenon. The co-administration of EUG and ML385 exhibited a comparatively attenuated effect (*Figure 8B*). The result of γH2AX cell immunofluorescence staining (*Figure 8C*) was in accordance with the findings observed through western blot (*Figure 8D*). Occurrence of oxidative stress can induce cellular apoptosis (*Liu et al., 2021*). The expressions of apoptosis-related proteins, including BCL2, BAX, and Cleaved Caspase-3, were examined using western blot (*Figure 9A*). The

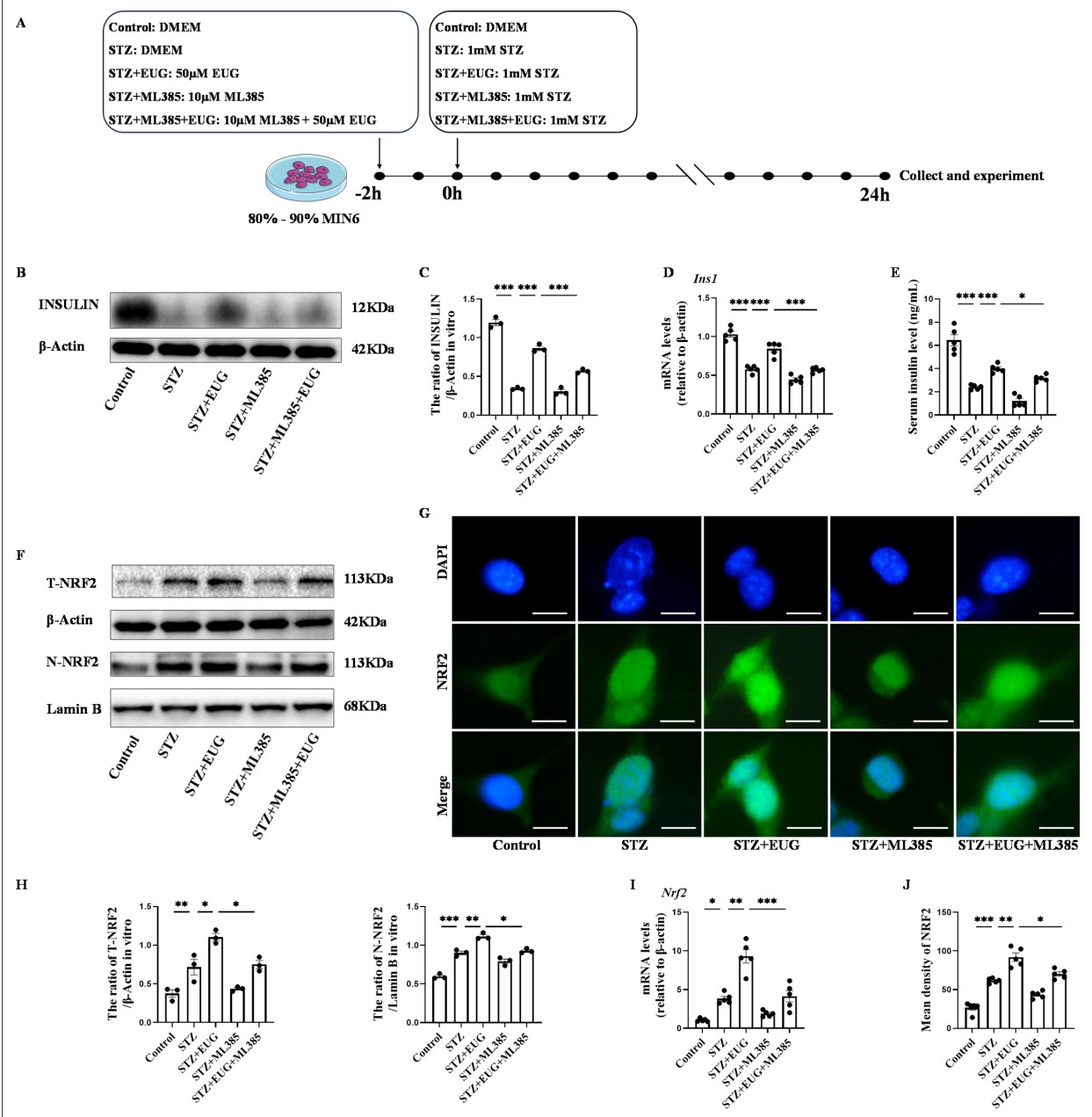

**Figure 6.** Eugenol (EUG) improved streptozotocin (STZ)-induced MIN6 cells insulin secretion by facilitating nuclear factor E2-related factor 2 (NRF2) nuclear translocation in vitro. (**A**) The schematic diagram depicts the different interventions in cell experiments in MIN6 cells. (**B**) The detection of insulin expression in different groups using western blot. (**C**) The quantification of western blot gel bands in different groups (n=3 independently repeated experiments). (**D**) The gene levels of *Ins1* in different groups (n=5 independently repeated experiments). (**E**) Enzyme-linked immunosorbent assay (ELISA) analysis of serum insulin levels of MIN6 cell in different groups (n=5 independently repeated experiments). (**F**) The detection of T-NRF2, N-NRF2 expression in different groups using western blot. (**G**) The representative immunofluorescence staining images of NRF2 (green) in each group of MIN6 cells. Nuclei were stained with DAPI (blue). Scale bar 10 μm. (**H**) The quantification of western blot gel bands in different groups (n=3 independently repeated experiments). (**I**) The gene levels of *Nrf2* in different groups (n=5 independently repeated experiments). (**J**) The quantification of immunofluorescence staining in different groups (n=5 independently repeated experiments). Mean ± SEM. *p<0.05, **p<0.01, ***p<0.001 indicate significant differences, and ns>0.05 means no significance difference. One-way ANOVA.

The online version of this article includes the following source data and figure supplement(s) for figure 6:

**Source data 1.** Raw unedited gels for *Figure 6*.

**Source data 2.** Uncropped and labeled gels for *Figure 6*.

**Figure supplement 1.** Explored the optimal concentration of drugs for MIN6 cells in vitro.

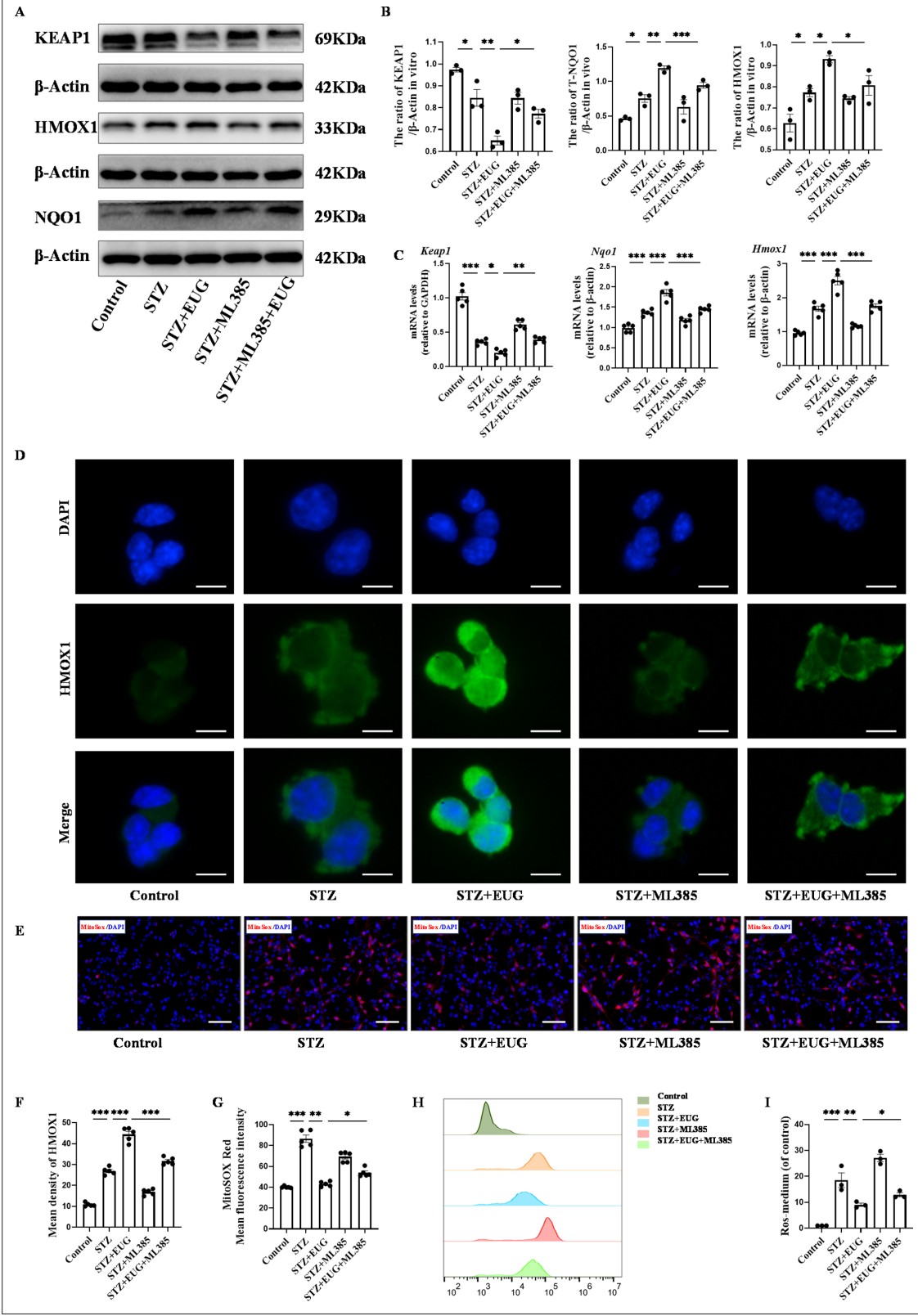

**Figure 7.** Eugenol (EUG) promoted the expression of nuclear factor E2-related factor 2 (NRF2) signaling pathway-related proteins to reduce intracellular reactive oxygen species (ROS) level. (**A**) The detection of KEAP1, HMOX1, and NQO1 expression in different groups using western blot. (**B**) The quantification of western blot gel bands in different groups (n=3 independently repeated experiments). (**C**) The gene levels of *Keap1*, *Nqo1*, and *Hmox1* in different groups (n=5 independently repeated experiments). (**D**) The representative immunofluorescence staining images of HMOX1 (green) in each

*Figure 7 continued on next page*

*Figure 7 continued*

group of MIN6 cells. Nuclei were stained with DAPI (blue). Scale bar 10 μm. (**E**) The generation of mitochondrial ROS in each group was detected by MitoSOX (red) and DAPI (blue) staining. Scale bar 100 μm. (**F**) The quantification of immunofluorescence staining in different groups (n=5 independently repeated experiments). (**G**) The quantitative analysis of immunofluorescence staining in different groups (n=5 independently repeated experiments). (**H**) The cell ROS in each group was analyzed using flow cytometry after DCFH-DA staining. (**I**) The quantitative analysis of flow cytometry after DCFH-DA staining (n=3 independently repeated experiments). Mean ± SEM. *p<0.05, **p<0.01, ***p<0.001 indicate significant differences, and ns>0.05 means no significance difference. One-way ANOVA.

The online version of this article includes the following source data for figure 7:

**Source data 1.** Raw unedited gels for *Figure 7*.

**Source data 2.** Uncropped and labeled gels for *Figure 7*.

results demonstrated an induction of apoptosis in STZ-induced MIN6 cells, while EUG intervention played an anti-apoptotic role and ML385 could reverse this phenomenon (*Figure 9B*). The RT-qPCR results of *Bcl2* and *Bax* were consistent with above findings (*Figure 9C*). Additionally, TUNEL staining (*Figure 9D*) and flow cytometry of apoptosis (*Figure 9F*) were performed, and the trends of results were also consistent with the aforementioned results (*Figure 9E and G*).

Based on the above findings, we hypothesized that EUG possesses the capability to ameliorate functional impairment of islet β cells in T1DM by reducing oxidative stress and apoptosis through activating the NRF2 signaling pathway. The potential mechanism of EUG in T1DM was illustrated in *Figure 10*.

## Discussion

The pathogenesis of T1DM involves the autoimmune destruction of pancreatic β cells, leading to an absolute deficiency in insulin secretion. Consequently, T1DM is more prevalent among adolescents (*Lu and Zhao, 2020*). Although subcutaneous insulin injection is the conventional clinical treatment for T1DM, it fails to improve life quality of patients. Therefore, there is an urgent to explore more efficacious therapies for T1DM. A large number of studies have demonstrated that oxidative stress

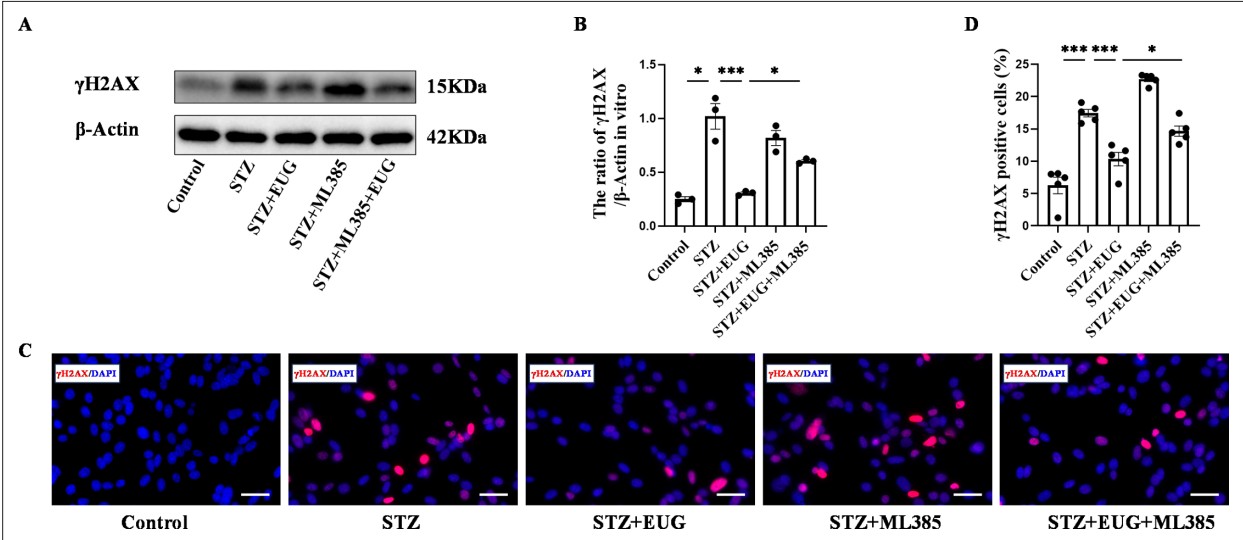

**Figure 8.** Eugenol (EUG) attenuated γH2AX expression in streptozotocin (STZ)-induced MIN6 cells in vitro. (**A**) The detection of γH2AX expression in different groups using western blot. (**B**) The quantification of western blot gel bands in different groups (n=3 independently repeated experiments). (**C**) The representative immunofluorescence staining images of γH2AX (red) in each group of MIN6 cells. Nuclei were stained with DAPI (blue). Scale bar 50 μm. (**D**) The quantitative analysis of γH2AX positive cells in different groups (n=5 independently repeated experiments). Mean ± SEM. *p<0.05, **p<0.01, ***p<0.001 indicate significant differences, and ns>0.05 means no significance difference. One-way ANOVA.

The online version of this article includes the following source data for figure 8:

**Source data 1.** Raw unedited gels for *Figure 8*.

**Source data 2.** Uncropped and labeled gels for *Figure 8*.

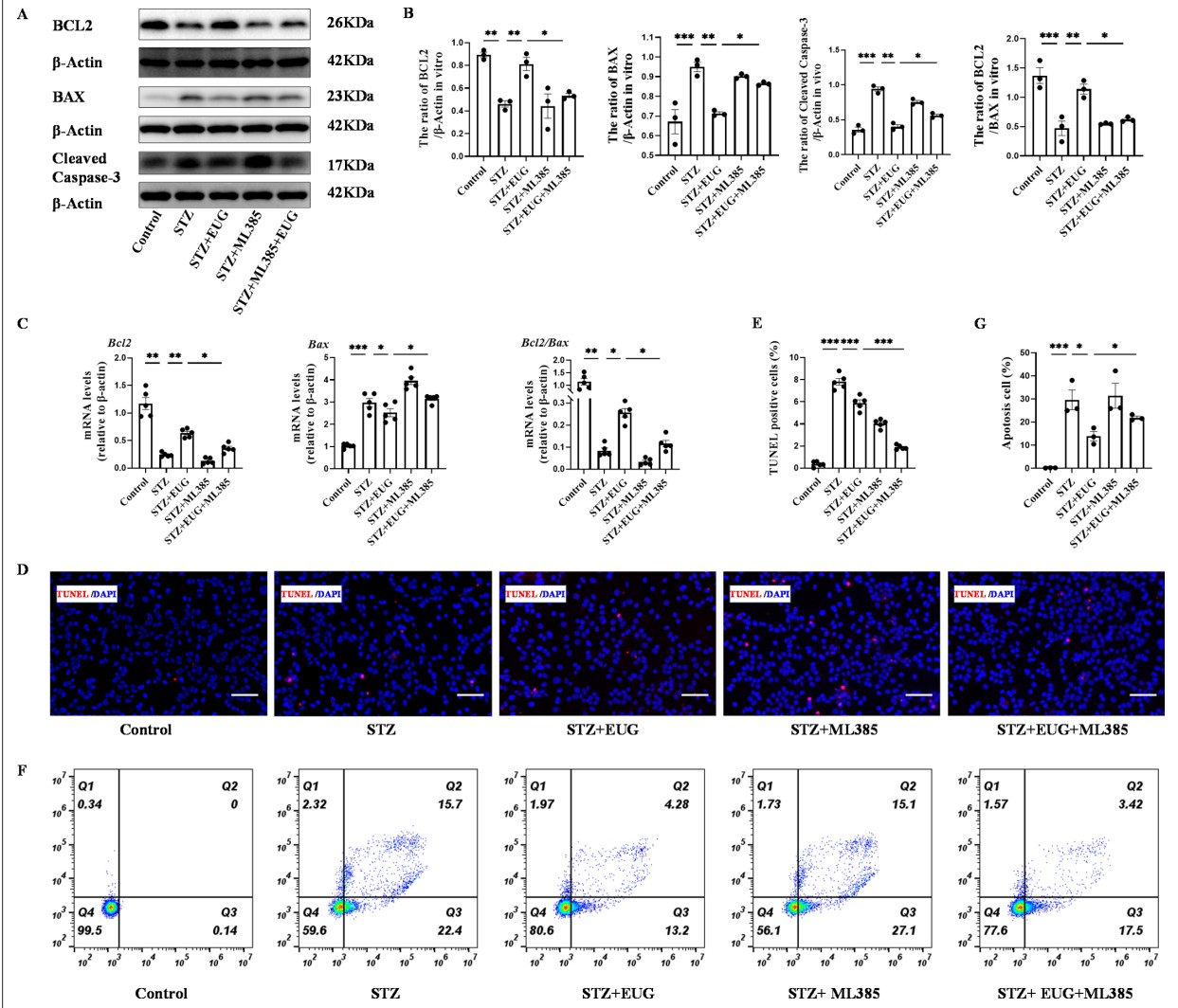

**Figure 9.** Eugenol (EUG) exerted protection of streptozotocin (STZ)-induced MIN6 cells through inhibition of the apoptosis in vitro. (**A**) The detection of BCL2, BAX, Cleaved Caspase-3 expression in different groups using western blot. (**B**) The quantification of western blot gel bands in different groups (n=3 independently repeated experiments). (**C**) The gene levels of *Bcl2*, *Bax*, and *Bcl2/Bax* in different groups (n=5 independently repeated experiments). (**D**) The detection of MIN6 cells apoptosis in each group using TUNEL staining. The cells with red fluorescence represent apoptosis. Scale bar 100 μm. (**E**) The quantitative analysis of TUNEL positive cells in different groups (n=5 independently repeated experiments). (**F**) The apoptosis in each group was analyzed using flow cytometry after Annexin V FITC and PI co-staining. (**G**) The quantitative analysis of flow cytometry after Annexin V FITC and PI co-staining in different groups (n=5 independently repeated experiments). Mean ± SEM. *p<0.05, **p<0.01, ***p<0.001 indicate significant differences, and ns>0.05 means no significance difference. One-way ANOVA.

The online version of this article includes the following source data for figure 9:

**Source data 1.** Raw unedited gels for *Figure 9*.

**Source data 2.** Uncropped and labeled gels for *Figure 9*.

and apoptosis play crucial roles in causing islet damage in T1DM (*Piganelli et al., 2020*; *Santin and Eizirik, 2013*). In this study, we aimed to investigate the protective effects of EUG on islet damage in both in vivo and in vitro models of T1DM, and tried to illuminate the underlying mechanism.

EUG is a phenolic compound, which is present in various aromatic plants, exhibits a wide range of pharmacological properties including anti-oxidant, anti-inflammatory, anti-cancer, and anti-bacterial activities (*Ulanowska and Olas, 2021*). The administration of EUG has been shown to effectively lower blood glucose levels, ameliorate insulin resistance (*Al-Trad et al., 2019*), inhibit the activity of key diabetes-related enzymes in diabetic rats (*Mnafgui et al., 2013*), and suppress the production of advanced glycosylation end products associated with diabetes (*Singh et al., 2016a*). Therefore,

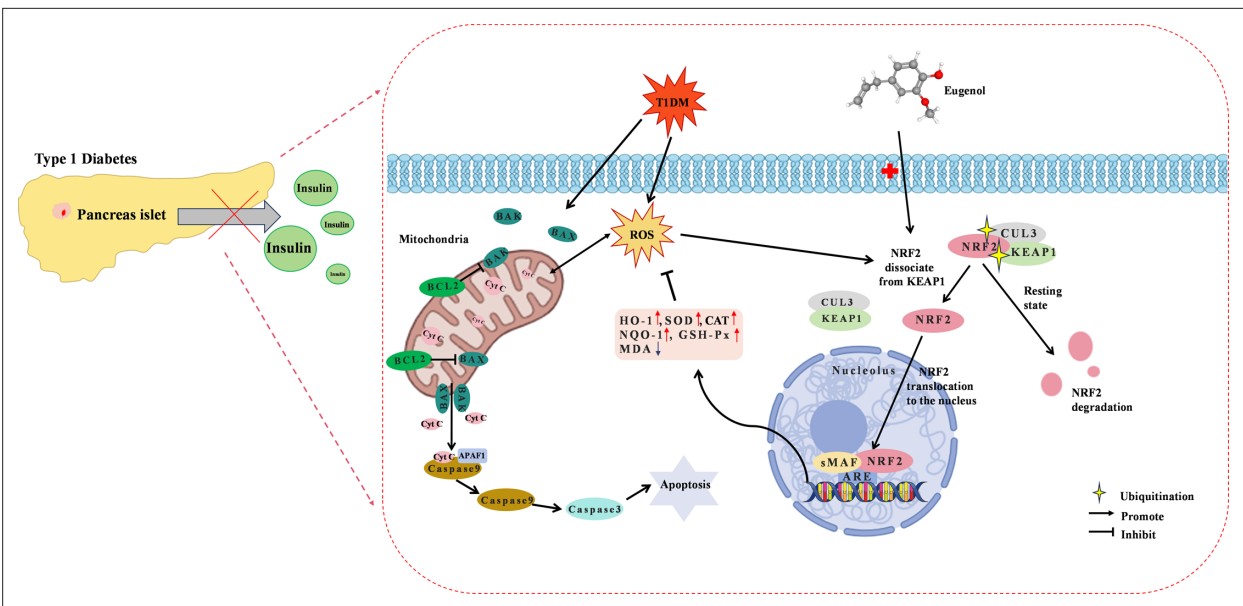

**Figure 10.** Diagram of the underlying mechanisms involved in the protective effects of eugenol (EUG) on type 1 diabetes mellitus (T1DM).

it is speculated that EUG may have a protective effect on diabetes. The meta-analysis conducted by *Carvalho et al., 2021* investigated the impact of EUG treatment on hyperglycemic murine models. The findings demonstrated that purified EUG effectively improved glucose levels, blood lipids, body weight, and restored anti-oxidant defense in diabetic animals. Moreover, EUG exhibited a reduction in markers associated with kidney damage in experimental diabetic animals. The purity of the EUG utilized in this study was 99.3%, and the results in this research were consistent with the meta-analysis in Carvalho et al. In this study, we will try to explore the protective effects of EUG on STZ-induced in vitro and in vivo T1DM models. Our in vivo results showed that administration of EUG could effectively ameliorate the T1DM symptoms such as polydipsia, polyphagia, polyuria, and weight loss. Additionally, it also alleviates ketonuria and glycosuria of T1DM mice. The islet damage in T1DM mice was mitigated, and insulin secretion was enhanced, along with effective alleviation of hyperglycemia symptoms, due to the intervention of EUG. As a chronic disease, diabetes can affect multiple organs, including the kidney, liver, and heart (*Papadopoulou-Marketou et al., 2018*; *Wang et al., 2022a*; *Khoury et al., 2018*). In this study, PAS staining showed that EUG intervention could reduce glomerular glycogen accumulation and improved the prognosis in T1DM mice. Our in vitro data showed that treatment with EUG could ameliorate the functional impairment of MIN6 cells induced by STZ, leading to increased insulin secretion levels, and reduced apoptosis and ROS production. Therefore, it can be speculated that EUG has the potential to treat T1DM.

The individuals suffering from diabetes exhibit persistent hyperglycemia, resulting in the generation of ROS that can cause cellular damage through various mechanisms (*Chen et al., 2018*). Mitochondrial function serves as the primary source of ROS, and in cases of mitochondrial dysfunction, there is an augmented production of ROS within the mitochondrial respiratory chain (*Pieczenik and Neustadt, 2007*). The mitochondrial respiratory chain complex enzymes are impaired under persistent hyperglycemia, leading to the development of secondary complications in diabetes (*Kowaltowski et al., 2009*; *Jezek and Hlavatá, 2005*). Therefore, oxidative stress is widely recognized as a major factor in the pathogenesis of diabetes. Our RNA-seq analyses showed that the oxidative stress signaling were highly activated in T1DM mice and improved after intervention with EUG. Similarly, our study showed an elevation in MDA levels in T1DM mice. This biomarker is associated with oxidative stress that represents the end product of lipid peroxidation caused by free radicals (*Ayala et al., 2014*). Additionally, anti-oxidant enzymes SOD, CAT, and GSH-Px can facilitate the catabolism of superoxide and peroxides in order to protect the body from oxidative stress damage. In our study, there was a reduction in the levels of SOD, CAT, and GSH-Px in T1DM group in vivo, while intervention with EUG could inhibit MDA level and promote productions of SOD, CAT, and GSH-Px in T1DM mice. These anti-oxidant enzymes subsequently inhibited the oxidative stress response in T1DM mice by

restricting the occurrence of free radical reactions. Mitochondria play a pivotal role in the regulation of oxidative stress. The results of MitoSOX fluorescence staining and ROS flow cytometry in vitro results showed that there was a increase in mitochondrial ROS and intracellular total ROS levels in STZ-induced MIN6 cells, which could be attenuated by EUG intervention. Furthermore, both in vivo and in vitro findings have demonstrated that EUG exhibited the ability to augment the expression of the relevant indicators within the NRF2 signaling pathway, which is associated with anti-oxidant stress response. To further verify the association between EUG and the NRF2 signaling pathway, the NRF2-specific inhibitor ML385 was used in vitro experiments (*Singh et al., 2016b*). Notably, ML385 could abolish the effects of EUG. Therefore, we hypothesized that the activation of NRF2 signaling pathway might function as a protective mechanism of EUG against STZ-induced T1DM.

NRF2 is a transcription factor that plays a crucial role in cellular response to oxidative stress by inducing the expression of various anti-oxidants (*Chen and Maltagliati, 2018*), and its activity is regulated by the endogenous inhibitor KEAP1 (*Kansanen et al., 2013*). The NRF2 and KEAP1 proteins form a conserved intracellular defense mechanism to combat oxidative stress. Normally, KEAP1-CUL3-E3 ubiquitin ligase typically targets the N-terminal Neh2 domain of NRF2, and then facilitates NRF2 ubiquitination, thereby maintaining a low level of NRF2 (*Wang et al., 2020*). In response to oxidative stress, specific cysteine residues in KEAP1 are modified that induce conformational changes in the KEAP1-CUL3-E3 ubiquitin ligase complex, thereby disrupting NRF2 ubiquitination and facilitating its translocation into the nucleus for heterodimerization with small MAF protein (sMAF). This complex subsequently binds to ARE located in the promoter region of various cytoprotective genes, thereby facilitating their transcription and inducing the expression of a range of cell-protective genes, including NQO1, HMOX1, SOD, CAT, GSH-Px. This process plays a crucial role in mitigating oxidative stress, exerting an anti-oxidative stress role (*Tonelli et al., 2018*). Previous study had demonstrated that EUG could attenuate the oxidative stress and apoptosis induced by transmissible gastroenteritis virus through the ROS-NRF2-ARE signaling pathway (*Wang et al., 2022b*). To further explore the potential molecular mechanisms underlying the regulation in T1DM, we conducted RNA-seq analysis on pancreatic tissues obtained from the Control group, T1DM group, and EUG group. Our findings from DEGs and heat map revealed the intervention of EUG could enhance the functionality of islet β cells, augment insulin secretion, and mitigate hyperglycemia in T1DM mice. GSEA showed that compared to the T1DM group, the EUG intervention might alleviate T1DM by improving oxidative stress pathway. Additionally, in this study, both in vivo and in vitro data showed that EUG activated NRF2, promoting its translocation into the nucleus, and inducing the expression of cytoprotective proteins NQO1 and HMOX1. Moreover, in vivo study showed that NRF2 regulated the levels of anti-oxidant enzymes such as SOD, CAT, and GSH-Px to suppress free radical reactions, exerting a protective effect on T1DM mice. Our in vivo results suggested that EUG has the potential to alleviate pancreatic β cell damage by activating the NRF2 pathway, thereby augmenting insulin secretion and ameliorating the prognosis of T1DM. Meanwhile, the intervention of ML385 counteracted the anti-oxidant and anti-apoptotic effects of EUG, leading to a reduction in NRF2 nuclear translocation and down-regulation of anti-oxidant oxidases. Therefore, we hypothesized the activation of the NRF2 signaling pathway through EUG might potentially alleviate damage of pancreatic β cells in T1DM.

Mitochondria are the main sites for cellular production of ROS, and intracellular ROS can function as signaling molecules to regulate physiological functions within the body. Accumulation of ROS caused by hyperglycemia can result in a reduction in mitochondrial membrane potential and an increased in membrane permeability (*Chen et al., 2005*; *Lepore et al., 2004*; *Rachek et al., 2006*). The apoptotic factor cytochrome *c* is released into the cytoplasm to initiate the activation of the Caspase cascade (e.g. Caspase9 and Caspase3 are activated successively), leading to chromosome aggregation and DNA fragmentation (*D'Amelio et al., 2010*). The protein Caspase-3, which is essential for cellular function, plays a pivotal role in the intricate process of programmed cell death known as apoptosis. The BCL-2 family proteins, encompassing anti-apoptotic proteins like BCL-2 and BCL-xl, as well as pro-apoptotic proteins such as BAX and BAD, can regulate this process (*Adams and Cory, 1998*). The dysregulation of the BCL2/BAX, characterized by aberrant expression of anti-apoptotic or pro-apoptotic genes, can initiate apoptosis and ultimately result in organ damage (*Liao et al., 2019*). The development of T1DM is attributed to the autoimmune destruction of islet β cells, and islet activation stimulates antigen-presenting cells, triggering the activation of CD4[+] helper T cells and subsequent the release of chemokines/cytokines (*Tsai et al., 2008*; *James et al., 2020*). The

activation of CD8[+] cytotoxic T cells is subsequently induced by these cytokines, resulting in the subsequent destruction of β cells. The apoptotic pathways of T1DM include exogenous (receptor-mediated) and endogenous (mitochondria-driven) mechanisms. Exogenous pathway encompasses CD4[+]-CD8[+] interacting Fas pathway, while endogenous pathway is characterized by a delicate balance between anti-apoptotic B-cell lymphoma (BCL-2) and BCL-xL proteins with pro-apoptotic Bax, Bad, Bid, and Bik proteins (*Tomita, 2017*). Therefore, targeted intervention of apoptosis can effectively enhance the prognosis of T1DM. The EUG compound has been reported to alleviate precancerous breast lesions by inhibiting apoptosis through the HER2/PI3K-AKT pathway (*Abdullah et al., 2021*). In this study, our in vivo and in vitro results showed that EUG treatment down-regulated the expression of BAX and Cleaved Caspase-3, while up-regulated the level of BCL2 in the T1DM mice. Additionally, it reduced the number of TUNEL positive cells, thereby exerting a protective effect on islet β cells in T1DM mice. The survival of cell is dependent on the integrity of DNA, as any damage to DNA has the potential to induce cellular apoptosis or necrosis (*Kinner et al., 2008*). In this study, our data demonstrated that the expression of the DNA damage biomarker γH2AX was elevated in T1DM group both in vivo and in vitro, whereas intervention with EUG could reversed this trend. The involvement of NRF2 in the regulation of apoptosis has been observed, and cells lacking NRF2 exhibit an increased occurrence of spontaneous apoptosis (*Merchant et al., 2011*). Our in vitro results showed that the NRF2-specific inhibitor ML385 could attenuate the anti-apoptosis effect of EUG on STZ-induced MIN6 cells. Consequently, we speculated that EUG exerts its apoptotic suppression through activating the NRF2 signaling pathway. The findings of this study suggest that EUG may exert a protective effect on pancreatic β cell damage in T1DM by mitigating oxidative stress and apoptosis through the activation of the NRF2 signaling pathway.

In this study, in vivo and in vitro results suggested that EUG may exert a protective effect on pancreatic β cell damages in T1DM by alleviating oxidative stress and apoptosis via activating the NRF2 signaling pathway. However, there are certain limitations to our study. First, ML385 was solely used to assess the protective effect of EUG in vitro but not in vivo, and the inhibitory effect on NRF2 in vivo needs to be further investigated. Second, it is imperative to analyze the dynamics of NRF2 decay in order to ascertain whether EUG affects the stability of NRF2 protein and gain a deeper understanding of its mechanism of action. Lastly, although MIN6 cells are extensively utilized in diabetes in vitro research, the primary islet cells would be optimal for studying T1DM in vitro.

In conclusion, our studies have demonstrated that EUG treatment could alleviate the symptoms associated with T1DM and ameliorate the damage of islets in T1DM mice. Moreover, EUG exhibited the ability to attenuate the deleterious effects on MIN6 cells induced by STZ in vitro. Additionally, EUG exhibits inhibitory effects on the oxidative stress and apoptosis in T1DM group in vivo and in vitro by activating the NRF2-mediated oxidative stress pathway. These findings suggested that EUG holds potential as a therapeutic option for patients with T1DM.

## Materials and methods

**Key resources table**

| Reagent type (species) or resource | Designation | Source or reference | Identifiers | Additional information |
|---|---|---|---|---|
| Cell line (*Mus musculus*, mouse) | MIN6 | GuangZhou Jennio Biotech Co., Ltd | Cat# JNO-M0060 RRID:CVCL_0431 | |
| Antibody | Anti-Insulin (Rabbit monoclonal) | Abcam | Cat# ab181547, RRID:AB_2716761 | IF/IHC (1:200), WB (1:1000) |
| Antibody | Anti-Phospho-Histone H2A.X (Ser139) (Rabbit polyclonal) | Abmart | Cat# T56572, RRID:AB_2936396 | IF/IHC (1:100), WB (1:5000) |
| Antibody | Anti-BAX (Mouse monoclonal) | Santa Cruz Biotechnology | Cat# sc-7480, RRID:AB_626729 | WB (1:500) |
| Antibody | Anti-Cleaved Caspase-3 (Rabbit monoclonal) | Cell Signaling Technology | Cat# 9664, RRID:AB_2070042 | WB (1:1000) |

*Continued on next page*

*Continued*

| Reagent type (species) or resource | Designation | Source or reference | Identifiers | Additional information |
|---|---|---|---|---|
| Antibody | Anti-NRF2 (Mouse monoclonal) | Proteintech | Cat# 16396-1-AP; RRID:AB_2782956 | IF (1:200), WB (1:1000) |
| Antibody | Anti-KEAP1 (Mouse monoclonal) | Proteintech | Cat# 60027-1-Ig, RRID:AB_2132623 | WB (1:1000) |
| Antibody | Anti-HO-1/HMOX1 (Rabbit polyclonal) | Proteintech | Cat# 10701-1-AP, RRID:AB_2118685 | IF (1:200), WB (1:1000) |
| Antibody | Anti-NQO1 (Mouse monoclonal) | Proteintech | Cat# 67240-1-Ig, RRID:AB_2882519 | WB (1:1000) |
| Antibody | Anti-β-actin (Rabbit polyclonal) | Proteintech | Cat# 20536-1-AP, RRID:AB_10700003 | WB (1:5000) |
| Antibody | Anti-Lamin B (Rabbit polyclonal) | Proteintech | Cat# 12987-1-AP, RRID:AB_2136290 | WB (1:1000) |
| Antibody | Anti-Bcl-2 (Mouse monoclonal) | Affinity Biosciences | Cat# BF9103, RRID:AB_2837570 | WB (1:1000) |
| Antibody | Anti-GAPDH (Rabbit polyclonal) | Affinity Biosciences | Cat# AF7021, RRID:AB_2839421 | WB (1:5000) |
| Sequence-based reagent | Ins1_F (Mouse) | This paper | PCR primers | CAAACCCACCCAGGCTTTTG |
| Sequence-based reagent | Ins1_R (Mouse) | This paper | PCR primers | AACGCCAAGGTCTGAAGGTC |
| Sequence-based reagent | Bax_F (Mouse) | This paper | PCR primers | ACACTGGACTTCCTCCGTGA |
| Sequence-based reagent | Bax_R (Mouse) | This paper | PCR primers | AGAGGAGGCCTTCCCAGC |
| Sequence-based reagent | Bcl2_F (Mouse) | This paper | PCR primers | TGAACTGGGGGAGGATTGTG |
| Sequence-based reagent | Bcl2_R (Mouse) | This paper | PCR primers | CAGAGACAGCCAGGAGAAATCA |
| Sequence-based reagent | Nrf2_F (Mouse) | This paper | PCR primers | CAGCCATGACTGATTTAAGCAG |
| Sequence-based reagent | Nrf2_R (Mouse) | This paper | PCR primers | CAGCTGCTTGTTTTCGGTATTA |
| Sequence-based reagent | HMOX1_F (Mouse) | This paper | PCR primers | TCCTTGTACCATATCTACACGG |
| Sequence-based reagent | HMOX1_R (Mouse) | This paper | PCR primers | GAGACGCTTTACATAGTGCTGT |
| Sequence-based reagent | Keap1_F (Mouse) | This paper | PCR primers | GACTGGGTCAAATACGACTGC |
| Sequence-based reagent | Keap1_R (Mouse) | This paper | PCR primers | GAATATCTGCACCAGGTAGTCC |
| Sequence-based reagent | NQO1_F (Mouse) | This paper | PCR primers | GAAGACATCATTCAACTACGCC |
| Sequence-based reagent | NQO1_R (Mouse) | This paper | PCR primers | GAGATGACTCGGAAGGATACTG |
| Sequence-based reagent | β-actin_F (Mouse) | This paper | PCR primers | CTACCTCATGAAGATCCTGACC |
| Sequence-based reagent | β-actin_R (Mouse) | This paper | PCR primers | CACAGCTTCTCTTTGATGTCAC |

*Continued on next page*

*Continued*

| Reagent type (species) or resource | Designation | Source or reference | Identifiers | Additional information |
|---|---|---|---|---|
| Commercial assay or kit | Mouse insulin (INS) ELISA Kit | ShangHai Boyun Biotech Co., Ltd | Cat# BP-E20353 | Mouse insulin (INS) ELISA Kit |
| Commercial assay or kit | Mouse ketonuria levels ELISA Kit | MEIMIAN | Cat# MM-0967M1 | Mouse ketonuria levels ELISA Kit |
| Commercial assay or kit | Urine Glucose Assay Kit | Nanjing Jiancheng Bioengineering Institute | Cat# C041-1-1 | Urine Glucose Assay Kit |
| Commercial assay or kit | Nuclear protein extraction kit | Beyotime Biotechnology | Cat# P0027 | Nuclear protein extraction kit |
| Commercial assay or kit | In Situ Cell Death Detection Kit, POD | Roche Life Science | Cat# 11684817910 | In Situ Cell Death Detection Kit, POD |
| Commercial assay or kit | In Situ Cell Death Detection Kit, Fluorescein | Roche Life Science | Cat# 11684795910 | In Situ Cell Death Detection Kit, Fluorescein |
| Commercial assay or kit | FITC Annexin V Apoptosis Detection Kit I | Becton Dickinson | Cat# 556547 | Annexin V FITC/PI apoptosis detection kit |
| Chemical compound, drug | Streptozotocin | Sigma-Aldrich | Cat# S0130-1G | |
| Chemical compound, drug | Eugenol | MedChemExpress | Cat# HY-N0337 | |
| Chemical compound, drug | Collagenase V solution | Sigma-Aldrich | Cat# C9263 | |
| Chemical compound, drug | Penicillin/streptomycin | Gibco | Cat# 15140122 | |
| Chemical compound, drug | Fetal bovine serum | Gibco | Cat# A3160801 | |
| Chemical compound, drug | MEM NON-ESSENTIAL AMINO ACIDS SOLUTION | Gibco | Cat# 11140050 | |
| Chemical compound, drug | ML385 | MedChemExpress | Cat# HY-100523 | |
| Chemical compound, drug | MitoSOX Red mitochondrial superoxide indicator | Yeasen | Cat# 50102ES02 | |
| Software, algorithm | ImageJ | ImageJ, https://imagej.nih.gov/ij/ | RRID:SCR_003070 | |
| Software, algorithm | GraphPad Prism | GraphPad Software, https://www.graphpad.com/ | RRID:SCR_002798 | |
| Software, algorithm | FlowJo | FlowJo Software, https://www.flowjo.com/solutions/flowjo | RRID:SCR_008520 | |

## Animal

The male C57BL/6 mice (n=150) weighing 18–20 g and aged 5–6 weeks were procured from Zhejiang Weitong Lihua Laboratory Animal Technology Co., LTD. The mice were housed in a temperature-controlled pathogen-free facility (SPF) environment with a light/dark cycle for 12 hr, maintained at a temperature of 23°C ± 2°C, relative humidity between 45% and 55%, and provided ad libitum access to food and water. All experimental operations were approved by the Ethics Committee of Laboratory Animals of Wenzhou Medical University, and were strictly adhered to the Guide for the Care and Use of Laboratory Animals (wydw2022-0208).

## Animal experiment

After 1 week of adaptive feeding, T1DM mouse model was established in C57BL/6 mice using STZ (Sigma-Aldrich, CA, USA). Briefly, each morning after 5 hr fasting period in each group of mice, STZ was dissolved in a 0.1 M sodium citrate buffer (Solarbio, China, Beijing), and then intraperitoneally injected into mice at a dose of 50 mg/kg for 5 consecutive days (*Furman, 2021*). The STZ solution was prepared under dark conditions and utilized promptly due to its inherent instability and photosensitivity. The mice were administered a 10% glucose solution instead of water during T1DM modeling to prevent mortality resulting from transient hypoglycemia. Fasting blood glucose levels were measured 3 days after modeling, and mice with fasting glucose levels more than 250 mg/dL were included in the study.

The intervention treatment of T1DM mice is conducted through two methods: oral administration (*Zhao et al., 2021*) and oral gavage (*Xing et al., 2023*; *Sudirman et al., 2019*; *Yao et al., 2023*). Due to limited experimental conditions, it is not feasible to feed a single mouse in a single cage, which makes it challenging to precisely control the actual daily intervention dose for each mouse when using oral administration. To ensure that each mouse receives an intervention dose according to its weight and expected dosage, we employ a method of gavage. In addition, oral gavage is more convenient and easier to operate than oral administration.

In terms of dosage setting for EUG (purity >98%, MedChemExpress, NJ, USA), previous studies have demonstrated that oral gavage of EUG (20 mg/kg/day) is effective in improving hyperglycemic symptoms in mice for 15 weeks (*Jeong et al., 2014*). Additionally, studies have shown that oral gavage of EUG (5, 10 mg/kg/day) can cure visceral leishmaniasis in mice (*Charan Raja et al., 2021*). Based on these reports, the T1DM mice in our study were divided into five groups: Control (n=30), T1DM (n=30), T1DM+EUG (5 mg/kg/day) (n=30), T1DM+EUG (10 mg/kg/day) (n=30), and T1DM+EUG (20 mg/kg/day) (n=30). Each group received daily oral gavage for 8 weeks, while the Control group received an equivalent volume of normal saline. Fasting blood glucose levels and fasting weight were measured and recorded on a weekly basis. Finally, serum samples were collected through orbital blood collection technique, while pancreatic tissue was isolated for subsequent experiments.

## Fasting blood glucose, OGTT

The OGTT is widely regarded as the criterion for diagnosing T1DM, which is crucial methods for assessing pancreatic islet function (*Helminen et al., 2015*). Following a 14 hr fasting period, the fasting blood glucose levels were measured in each group mice, and tail vein blood samples were collected and analyzed using a glucometer. OGTT was performed by administering a 20% glucose solution intragastrically after the fasting period, and blood glucose levels were measured at 0 min, 30 min, 60 min, 90 min, and 120 min following the initial gavage administration.

## Food, water, urine volume, and urine glucose measurements

The changes in food and water intake for each group were recorded throughout the experiment by regulating the daily initial food intake (100 g) and water quantity (250 mL). After successful modeling of T1DM mice, the experimental animals were grouped based on the section Animal experiment as follows: Control, T1DM, T1DM+EUG (5 mg/kg/day), T1DM+EUG (10 mg/kg/day), and T1DM+EUG (20 mg/kg/day). To ensure consistency among groups, each group consisted of five mice and had equal amounts of diet (100 g), drinking water (250 mL), and environmental conditions for feeding. The urine volume of each cage was quantified by measuring the area of the bedding material saturated with urine. The urine samples from each group of mice were collected for the detection of urine glucose levels using the Urine Glucose Assay Kit (Jiancheng, Nanjing, Jiangsu, China).

## Islet isolation

The method described by *Xu et al., 2010*, was employed for the isolation of islets. First, the common bile duct was ligated, and a retrograde injection of collagenase V solution (0.8 mg/mL; Sigma-Aldrich, CA, USA) was performed through the common bile duct until the pancreas was completely filled and foliated. The pancreas was then digested in 5 mL of collagenase V solution at a concentration of the same for 15 min at 37°C until the majority of pancreatic tissues were digested into chylous and silt. To stop the process of digestion, 10 mL of 4°C Hank's solution was added and then centrifuged after gentle shaking. The centrifugation was immediately stopped upon reaching a speed of 2000 rpm.

The supernatant was discarded and the sediment was washed twice using Hank's balanced salt solution. Islet tissues were placed in 5 mL of 4°C Ficoll density gradient medium (1.119). After mixing, a sequence of slow drips consisting of 2 mL 4°C Ficoll density gradient medium (1.077) and 2 mL Hank's solution was performed before centrifugation at a speed of 2000 rpm for a duration of 5 min. The sediment was washed again with Hank's solution, and subsequently was picked under a dissecting microscope (Leica, Germany). The isolated islets were promptly utilized for subsequent experimental procedures.

## Insulin and ketonuria ELISA
After collecting the culture supernatant of MIN6 cells and mouse serum from each group, insulin levels in the samples were evaluated using a commercially available ELISA assay kit (Boyun, Shanghai, China) in accordance with the manufacturer's protocol. The urine samples from each group were collected for the quantification of ketonuria levels using a commercially available ELISA kit (Boyun, Shanghai, China).

## PAS staining
After dewaxing the paraffin sections, a PAS staining kit (Servicebio, Wuhan, China) was utilized to detect glomerular glycogen accumulation. These tissue sections were immersed in a 0.5% periodate solution for 15 min, and subsequently stained with Schiff reagent for 30 min in darkness. Finally, the nuclei of these tissue sections were stained using hematoxylin reagent. The images of paraffin sections were captured using an optical microscope (Nikon, Japan).

## H&E staining
The fresh pancreatic tissues from mice in each group were carefully separated and promptly fixed with 4% paraformaldehyde (PFA; Solarbio, Beijing, China). After dehydration for 24 hr, they were embedded in paraffin and sliced into 5 μm. Subsequently, the paraffin sections of pancreas were stained with H&E staining kit (Servicebio, Wuhan, China), and the pathological changes of pancreatic islets were observed under an optical microscope.

## Immunohistochemistry staining
After dewaxing, the peroxidase blockade agent (Zsbio, Beijing, China) was applied to the tissue surface at room temperature (RT) for 20 min. Following a 15 min wash with PBS, sections were subjected to antigen repair by boiling in a pressure cooker containing 10 mM citrate acid buffer for 2 min (pH 6.0, Solarbio, Beijing, China). Subsequently, antigen blocking was performed using 10% goat serum (Beyotime, Shanghai, China), and then the slides were incubated with primary antibodies (listed in *Supplementary file 1*) overnight at 4°C. On the second day, these sections were incubated with goat anti-rabbit secondary antibody (1:2000, Affinity Biosciences) at 37°C for 2 hr, and then stained with a solution of 3,3'-diaminobenzidine (DAB) (Zsbio, Beijing, China) and hematoxylin. The images of pancreatic paraffin sections were captured using an optical microscope.

## RNA-sequencing
Total RNAs were isolated using the TRIzol Reagent (Invitrogen Life Technologies), after which the concentration, quality, and integrity were determined using a NanoDrop spectrophotometer (Thermo Fisher Scientific, MA, USA). Products were purified (AMPure XP system) and quantified using the Agilent high-sensitivity DNA assay on a Bioanalyzer 2100 system (Agilent). The sequencing library was then sequenced on NovaSeq 6000 platform (Illumina) by Shanghai Personal Biotechnology Co. Ltd.

The DESeq package was used to analyze the differences in gene expression profiles, and the gseGO function in the clusterProfiler package was used to conduct GSEA according to the Gene Ontology (GO) database. The analysis was performed in the R language environment (version 4.2.0). The gene expression data of the T1DM group and EUG intervention group were thoroughly preprocessed before analysis, including standardization, missing value processing, and other necessary quality control steps. The pre-processed data were used to assess the enrichment of gene sets associated with GO in different biological states. The GO database used covers three main domains: BP, MF, and cell components. The significance level was set as $p < 0.05$ and NES score $> 1.4$. The significance was corrected by multiple tests, and the p-value was adjusted by false discovery rate method. The

gseGO function was used to evaluate the enrichment of a gene set in a given gene expression data, including calculating the enrichment score (ES) and standardized enrichment score (NES) of the gene set.

## Cell culture and treatment

The mouse pancreatic β cell line MIN6 was obtained from the American Type Culture Collection (ATCC, Manassas, VA, USA), and was cultured in Dulbecco's Modified Eagle Medium (DMEM; Gibco, United States) supplemented with 1% penicillin/streptomycin (Gibco, CA, USA), 10% fetal bovine serum (Gibco, CA, USA), and 1% MEM NON-ESSENTIAL AMINO ACIDS SOLUTION (100×; NEAA, Gibco, USA) at a temperature of 37°C in a $CO_2$ incubator with a concentration of 5%. The cells were passaged every 3 days.

The in vitro cell model of T1DM was induced by STZ. MIN6 cells were treated with various concentrations of STZ (0.5 mM, 1 mM, 2 mM, 4 mM, and 8 mM) for a duration of 24 hr to determine the optimal working concentration. Subsequently, STZ-induced MIN6 cells were exposed to different concentrations of EUG (50 µM, 100 µM, 200 µM, 400 µM, and 600 µM) for a duration of 2 hr to identify the optimal working concentration. Based on the optimal working concentrations of STZ and EUG respectively, we set different groups, including Control, STZ, STZ+EUG, STZ+ML385 (10 µM, MedChemExpress, NJ, USA), STZ+ML385+EUG. Pre-treatment with either EUG or ML385 occurred 2 hr prior to stimulation with STZ, simultaneous administration of ML385 and EUG mixture took place at the same time. Finally, the cells or cell supernatants from each respective group were collected for subsequent experiments.

## Cell viability assay

The cell viability of MIN6 cells was assessed using the cell counting kit-8 (CCK-8, Yeasen, Shanghai, China) under different treatment conditions. MIN6 cells were seeded into 96-well plates at a density of $5×10^3$/well. Once the cells reached 80–90% confluence, they were subjected to treatment with different concentrations of drugs. The different doses of EUG were added into the 96-well plates 2 hr prior to STZ treatment. Subsequent to drug administration, each well was supplemented with 100 µL DMEM containing 10 µL CCK-8 solution. After incubation at 37°C for 30 min, the absorbance at 450 nm in each well was quantified using a microplate reader (Thermo Fisher Scientific, MA, USA).

## Western blot

The total proteins from mouse islets or MIN6 cells were obtained using RIPA lysis buffer (Solarbio, Beijing, China) supplemented with a phosphatase inhibitor (Solarbio, Beijing, China) and a serine protease inhibitor (Solarbio, Beijing, China). Nuclear protein extraction was performed using the nuclear protein extraction kit (Beyotime, Shanghai, China), following the manufacturer's instructions. Protein concentration was determined using the BCA protein assay kit (Beyotime, Shanghai, China). The samples containing 20–80 µg of proteins were separated on 7.5% or 12.5% sodium dodecyl sulfate-polyacrylamide gels, and subsequently transferred onto the polyvinylidene fluoride membranes (Thermo Fisher Scientific, MA USA). These membranes were then incubated with 8% skim milk (Beyotime, Shanghai, China) at RT for 3 hr. After being washed with Tris-buffered saline containing Tween-20 (TBST), these membranes were incubated overnight at 4°C with the primary antibodies listed in *Supplementary file 1*. The membranes were then incubated with goat anti-mouse or goat anti-rabbit IgG HRP secondary antibodies (Affinity, Melbourne, Australia) diluted at a ratio of 1:5000 and kept at RT for 3 hr. Finally, the protein bands were detected using the ECL chromogenic kit (EpiZyme, Shanghai, China), and visualized using the ChemiDic XRS imaging system (Bio-Rad, CA USA). The intensity of protein bands was analyzed using ImageJ software (National Institutes of Health, MD, USA), which was normalized to β-actin or lamin B band.

## Real-time quantitative polymerase chain reaction

Total RNAs were extracted from mouse islets or MIN6 cells using TRIzol reagent (Invitrogen, CA, USA). After measuring of RNA concentration by reading the OD value at 260 nm, cDNAs were synthesized from the RNAs using cDNA Synthesis SuperMix (TaKaRa, Kusatsu, Japan). RT-qPCR was then performed using SYBR Green SuperMix (TOYOBO, Osaka, Japan). Subsequently, cycle threshold (Ct)

values were collected and normalized to *β-actin* levels. The mRNA levels were calculated using the $2^{(-\Delta\Delta Ct)}$ method. The primer sequences are provided in *Supplementary file 2*.

## Immunofluorescence staining

MIN6 cells in different groups were fixed with 4% PFA for 20 min. Then, they were incubated with 0.3% Triton X-100 (Sigma-Aldrich, CA, USA) for 1 hr, followed by incubation with 10% goat serum. After rinsing with PBS three times, the cells were incubated overnight at 4°C with the primary antibody (listed in *Supplementary file 1*). Then, they were incubated with FITC-conjugated goat anti-rabbit IgG secondary antibody (1:100; EarthOx, LA, USA) for 2 hr at RT. Finally, the nuclei were stained using DAPI-containing anti-fluorescence quencher and observed under a fluorescence microscope (Nikon, Tokyo, Japan).

## TUNEL staining

In vivo experiments, pancreatic paraffin sections were initially dewaxed in a 60°C oven, and then were incubated with protease K solution at 37°C for 30 min. Following PBS rinsing, the sections were stained with TUNEL mixture (Roche, Basel, Switzerland) at 37°C for 1 hr in darkness. Then, DAB staining and hematoxylin staining were performed. Finally, the images of these sections were captured under an optical microscope.

In vitro experiments, the apoptosis of MIN6 cells in each group was assessed using the TUNEL apoptosis detection kit (Roche, Basel, Switzerland). Briefly, the cells were incubated with TUNEL reagent for 1 hr at 37°C in dark environment, and subsequently washed with PBS before being stained with DAPI. Lastly, they were observed under a fluorescence microscope.

## Annexin V and PI assay

The Annexin V FITC/PI apoptosis detection kit (Becton Dickinson, NJ, USA) was utilized for the assessment of apoptosis of MIN6 cells in different groups. Following collection and double washing with 4°C PBS, the cells were re-suspended in Binding Buffer. Subsequently, these cells were stained at RT with a mixture of 5 µL FITC Annexin V and 5 µL PI for a duration of 15 min in darkness. Finally, the quantification of cell apoptosis ratio was detected using the Flow Cytometer (Beckman, CA, USA).

## Mitochondrial ROS detection

Mitochondrial ROS levels were evaluated using the MitoSOX Red mitochondrial superoxide indicator (Yeasen, Shanghai, China). The reagent was diluted with dimethyl sulfoxide (DMSO, Solarbio, Beijing, China). Subsequently, the MIN6 cells in each experimental group were incubated with MitoSOX reagent for 30 min at 37°C in dark environment. Finally, the nucleus was stained with an anti-fluorescence quencher containing DAPI, and images were captured under a fluorescence microscope.

## ROS detection

Intracellular levels of ROS were measured using the ROS assay kit (S0033S, Beyotime, Shanghai, China) on the basis of the manufacturer's instructions. In brief, MIN6 cells in each group were digested with trypsin which has no EDTA. Cells together with cell supernatants were incubated with diluted DCFH-DA solution at 37°C for 30 min in the dark. Finally labeled cells were detected by Flow Cytometer (Beckman Coulter, Breya, CA, USA).

## Statistical analysis

In this study, all data were expressed as mean ± SEM and analyzed by GraphPad Prism 9.0 (GraphPad Software Inc, CA, USA). All experiments were repeated at least three times independently. Statistical significance was analyzed by Student's t-test or one-way ANOVA followed by Turkey's multiple comparisons. The criterion for statistical significance was set at a p-value of less than 0.05.

# Additional information

## Funding

| Funder | Grant reference number | Author |
|---|---|---|
| Key Research and Development Program of Zhejiang Province | 2023C03018 | Xiaoling Guo |
| Natural Science Foundation of Zhejiang Province | LY24H020008 | Xiaoling Guo |
| Wenzhou Medical University | 604090352/640 | Xiaoling Guo |
| Natural Science Foundation of Zhejiang Province | LTGY23H03003 | Weiping Ji |

The funders had no role in study design, data collection and interpretation, or the decision to submit the work for publication.

## Author contributions

Yalan Jiang, Conceptualization, Resources, Data curation, Software, Formal analysis, Supervision, Investigation, Methodology, Writing – original draft, Project administration, Writing – review and editing; Pingping He, Investigation, Methodology, Project administration; Ke Sheng, Software, Formal analysis, Methodology, Project administration; Yongmiao Peng, Visualization, Project administration; Huilan Wu, Formal analysis, Investigation; Songwei Qian, Project administration; Weiping Ji, Funding acquisition; Xiaoling Guo, Supervision, Project administration, Writing – review and editing; Xiaoou Shan, Funding acquisition, Writing – review and editing

## Author ORCIDs

Xiaoling Guo ⓘ https://orcid.org/0000-0003-3153-6249
Xiaoou Shan ⓘ https://orcid.org/0009-0009-0487-9474

## Ethics

All experimental operations were approved by the Ethics Committee of Laboratory Animals of Wenzhou Medical University, and were strictly adhered to the Guide for the Care and Use of Laboratory Animals (wydw2022-0208).

Reviewer #2 (Public review): https://doi.org/10.7554/eLife.96600.3.sa1
Reviewer #3 (Public review): https://doi.org/10.7554/eLife.96600.3.sa2
Author response https://doi.org/10.7554/eLife.96600.3.sa3

# Additional files

## Supplementary files

Supplementary file 1. Antibodies. ND = not detected; WB = western blot; IHC: immunohistochemistry; IF: immunofluorescence.

Supplementary file 2. Primer information for mouse.

MDAR checklist

## Data availability

All data generated or analysed during this study are included in the manuscript and supporting files.

The following dataset was generated:

| Author(s) | Year | Dataset title | Dataset URL | Database and Identifier |
|---|---|---|---|---|
| Jiang Y, He P, Sheng K, Peng Y, Wu H, Qian S, Ji W, Guo X | 2024 | The protective roles of Eugenol on type 1 diabetes mellitus through NRF2 mediated oxidative stress pathway | https://www.ncbi.nlm. nih.gov/bioproject/ PRJNA1153736/ | NCBI BioProject, PRJNA1153736 |

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
