## [Editor Report · eLife Assessment]

This **useful** study partially succeeds in providing **solid** evidence in support of the therapeutic potential of the plant-derived compound eugenol for ameliorating symptoms associated with STZ-induced oxidative stress, identifying Nuclear factor E2-related factor (Nrf2) as a mediator of the effects induced by eugenol. Although the study provides interesting data, there remain concerns associated with the STZ model and the rather superficial mechanistic assessment.

---

## [Referee Report · Reviewer #2 (Public review)]

Summary:

In this manuscript, the authors consider the effects of eugenol (EUG), a plant-produced substance known to reduce oxidative stress in various cellular contexts via Nrf2, in alleviating the effects of streptozotocin (STZ), a known rodent beta cell toxin. They claim that EUG treatment would be useful for T1D therapy.

Strengths:

The experiments shown are sufficiently clear and rather convincing in documenting that eugenol can revert the effects of streptozotocin on animal physiology as well as beta cell oxidative stress and cell death via activation of Nrf2.

In the revised manuscript the authors corrected/explained most of the specific inconsistencies/mistakes pointed out.

However, they did not address the opening paragraph that points out major concerns. I summarize them below, together with some that were dealt with in their response but still remain unaddressed or not commented upon.

- STZ treatment cannot be used as a T1D model for the reasons I outlined in my previous letter. I would have been happy to see a response on that but they did not provide any. The manuscript is misleading in this important respect.

- Mechanistically, the manuscript remains at a rather superficial level. I highlighted some possibilities to enrich the manuscript but none was addressed even in the discussion.

(a) How is eugenol penetrating the cell, is there a receptor that could be potentially targeted?

(b) Are there intermediary proteins that convey the effect to the Nrf2/Keap1 complex or is eugenol directly disrupting their interaction?

(c) What are direct downstream Nrf2 effectors?

(d) Besides, streptozotocin is also a powerful DNA alkylating agent, are such effects relieved by eugenol?

- It is puzzling that all molecular analyses show a gradual reversion effect with increasing doses of eugenol but this gradual effect is apparently missing in many of the physiological parameters assessed in Figure 1, including the all-important OGTT assays. Can the authors interpret this? In the high eugenol group in the OGTT assays there is a group of mice that are clearly outliers. Most likely the STZ treatment for these mice was not efficient and their inclusion skews the results. Besides, it is important to assess differences among eugenol groups (one way ANOVA). The statistical tests provided are incomplete and sometimes not done correctly.

- Given that medical research is still heavily biased in favor of analyses in males and given that the authors have analyzed in Figure 1 a very large number of animals what are the results stratified by sex?

---

## [Referee Report · Reviewer #3 (Public review)]

Summary:

This study by Jiang et al. aims to establish the streptozotocin (STZ)-induced type 1 diabetes mellitus (T1DM) mouse model in vivo and the STZ-induced pancreatic β cell MIN6 cell model in vitro to explore the protective effects of Eugenol (EUG) on T1DM. The authors tried to elucidate the potential mechanism by which EUG inhibits the NRF2-mediated anti-oxidative stress pathway. Overall, this study is well executed with solid data, offering an intriguing report from animal studies for a potential new treatment strategy for T1DM.

Strengths:

In vivo efficacy study is comprehensive and solid. Given STZ-induced T1DM is a devastating and harsh model, the in vivo efficacy from this compound is really impressive.

---

## [Author Response]

The following is the authors’ response to the original reviews.

**Public Reviews:**

**Reviewer #1 (Public Review):**
SummaryType 1 diabetes mellitus (T1DM) progression is accelerated by oxidative stress and apoptosis. Eugenol (EUG) is a natural compound previously documented as anti-inflammatory, anti-oxidative, and anti-apoptotic. In this manuscript by Jiang et al., the authors study the effects of EUG on T1DM in MIN6 insulinoma cells and a mouse model of chemically induced T1DM. The authors show that EUG increases nuclear factor E2-related factor 2 (Nrf2) levels. This results in a reduction of pancreatic beta-cell damage, apoptosis, oxidative stress markers, and a recovery of insulin secretion. The authors highlight these effects as indicative of the therapeutic potential of EUG in managing T1DM.StrengthsRelevant, timely, and addresses an interesting question in the field. The authors consistently observe enhanced beta cell functionality following EUG treatment, which makes the compound a promising candidate for T1DM therapy.Weaknesses(1) The in vivo experiments have too few biological replicates. With an n=3 (as all figure legends indicate) in complex mouse studies such as these, drawing robust conclusions becomes challenging. It is important to reproduce these results in a larger cohort, to validate the conclusions of the authors.

Thanks for your comments. In the figure legends of the first draft manuscript, n=3 means at least 3 biological replicates, and in the section of material and methods, n=30 means sample size. The number of mice in each group is 30 and there were 150 mice used in this study, and mice are assigned as follows for the whole in vivo experiments. The relative information has been added in the revised manuscript.

(2) Another big concern is the lack of quantifications and statistical analysis throughout the manuscript. Although the authors claim statistical significance in various experiments, the limited information provided makes it difficult to verify. The authors use vague and minimal descriptions of their experiments, which further reduces the reader's comprehension and the reproducibility of the experiments.

Thanks for your constructive suggestion. We conducted quantitative and statistical analysis of the entire manuscript through GraphPad Prism software again. Additionally, we have improved the experimental description in the revised manuscript.

(3) Finally, the use of Min6 cells as a model for pancreatic beta cells is a strong limitation of this study. Future studies should seek to reproduce these findings in a more translational model and use more relevant in vitro cell systems (eg. Islets).

Thanks for your professional comments. Mouse insulinoma cells (MIN6 cell line) are permanent cell lines isolated from mouse islet β cell tumors, which can reflect the functional changes of islet β cells. As mature islet cells, MIN6 cells have been widely used in the study of type 1 diabetes mellitus[1-4], so in this study, MIN6 cells were used as the cell model in vitro. In our future studies, we will try to conduct our findings using more relevant in vitro cell systems (eg. Islets).

References:

(1) WU M, CHEN W, ZHANG S, et al. Rotenone protects against β-cell apoptosis and attenuates type 1 diabetes mellitus [J]. Apoptosis, 2019, 24(11-12): 879-91.

(2) LUO C, HOU C, YANG D, et al. Urolithin C alleviates pancreatic β-cell dysfunction in type 1 diabetes by activating Nrf2 signaling [J]. Nutr Diabetes, 2023, 13(1): 24.

(3) LAKHTER A J, PRATT R E, MOORE R E, et al. Beta cell extracellular vesicle miR-21-5p cargo is increased in response to inflammatory cytokines and serves as a biomarker of type 1 diabetes [J]. Diabetologia, 2018, 61(5): 1124-34.

(4) LIN Y, SUN Z. Antiaging Gene Klotho Attenuates Pancreatic β-Cell Apoptosis in Type 1 Diabetes [J]. Diabetes, 2015, 64(12): 4298-311.

**Reviewer #3 (Public Review):**
Summary:This study by Jiang et al. aims to establish the streptozotocin (STZ)-induced type 1 diabetes mellitus (T1DM) mouse model in vivo and the STZ-induced pancreatic β cell MIN6 cell model in vitro to explore the protective effects of Eugenol (EUG) on T1DM. The authors tried to elucidate the potential mechanism by which EUG inhibits the NRF2-mediated anti-oxidative stress pathway. Overall, this study is well executed with solid data, offering an intriguing report from animal studies for a potential new treatment strategy for T1DM.Strengths:The in vivo efficacy study is comprehensive and solid. Given that STZ-induced T1DM is a devastating and harsh model, the in vivo efficacy of this compound is really impressive.Weaknesses:(1) The Mechanism is linked with the anti-oxidant property of the compound, which is common for many natural compounds, such as flavonoids and polyphenol. However, rarely, this kind of compound has been successfully developed into therapeutics in clinical usage. Indeed, if that is the case, Vitamin C or Vitamin E could be used here as the positive control.

Thanks for your comments. In fact, many anti-oxidant drugs are used for the treatment of type 1 diabetes mellitus in the clinical. For example, lipoic acid was used to treat diabetic peripheral neuropathy[5]. Vitamin E could effectively eliminate free radicals, protect cell membranes, and significantly reduce the risk of cardiovascular disease in patients with SPACE or ICARE diabetes[6]. Glutathione played crucial roles in the detoxification and anti-oxidant systems of cells and has been used to treat acute poisoning and chronic liver diseases by intravenous injection[7]. Therefore, eugenol enhances the management of type 1 diabetes mellitus by modulating oxidative stress pathways and holds potential as a future therapeutic choice for clinical application. In the future relevant studies, we will try to use Vitamin C or Vitamin E as the positive control.

References:

(5) ZIEGLER D, PAPANAS N, SCHNELL O, et al. Current concepts in the management of diabetic polyneuropathy [J]. J Diabetes Investig, 2021, 12(4): 464-75.

(6) VARDI M, LEVY N S, LEVY A P. Vitamin E in the prevention of cardiovascular disease: the importance of proper patient selection [J]. J Lipid Res, 2013, 54(9): 2307-14.

(7) HONDA Y, KESSOKU T, SUMIDA Y, et al. Efficacy of glutathione for the treatment of nonalcoholic fatty liver disease: an open-label, single-arm, multicenter, pilot study [J]. BMC Gastroenterol, 2017, 17(1): 96.

**Reviewer #1 (Recommendations For The Authors):**
• For each of the figure panels the authors should indicate the exact number of biological replicates (how many mice or how many independent in vitro experiments). For IF panels, the number of mice, the number of histology slides per mouse, number of fields analyzed should be indicated.

Thanks for your constructive suggestion. These details had been added in the revised manuscript.

• The methods state n=30 and Figure 1 states n=3. N=3 is too little for such a complex in vivo study and would severely reduce the reliability of the in vivo experiments.

Thanks for your suggestion. In the figure legends of the first draft manuscript, n=3 means at least 3 biological replicates, and in the section of material and methods, n=30 means sample size. The number of mice in each group is 30 and there were 150 mice used in this study, and mice are assigned as follows for the whole in vivo experiments. The in vivo experimental data of Figure 1 were supplemented in the revised manuscript.

• Individual data points should be included in each of the graphs from this manuscript.

Thanks for your reminder. The revised manuscript have shown the individual data points in each of the graphs.

• The quantifications and statistics in the manuscript need improvement. Several experiments are missing quantifications and/or statistical tests (e.g. Figure 1J). Other experiments show a quantification but without any explanation of replicates (e.g. Figures 2B and 2G). None of the experiments show individual data points, and as in the previous comment, these should be included.

Thanks for your comments. In the revised manuscript, statistics and repetitions of experimental data have been supplemented, and individual data points were shown in each graph.

• What is the reason for intragastric administration? The previous studies on which the dosages were based used oral administration (gavage). (Discussed in methods 4.2).

Thanks for your professional comments. The intervention treatment of T1DM mice is conducted through two methods: oral administration[8] and oral gavage[9-11]. Due to limited experimental conditions, it is not feasible to feed a single mouse in a single cage, which makes it challenging to precisely control the actual daily intervention dose for each mouse when using oral administration. To ensure that each mouse receives an intervention dose according to its weight and expected dosage, we employ a method of gavage. In addition, oral gavage is more convenient and easier to operate than oral administration. Therefore, in vivo experiment of this study used eugenol gavage intervention as a treatment method. These details had been added in the revised manuscript.

References:

(8) ZHAO H, WU H, DUAN M, et al. Cinnamaldehyde Improves Metabolic Functions in Streptozotocin-Induced Diabetic Mice by Regulating Gut Microbiota [J]. Drug Des Devel Ther, 2021, 15: 2339-55.

(9) XING D, ZHOU Q, WANG Y, et al. Effects of Tauroursodeoxycholic Acid and 4-Phenylbutyric Acid on Selenium Distribution in Mice Model with Type 1 Diabetes [J]. Biol Trace Elem Res, 2023, 201(3): 1205-13.

(10) SUDIRMAN S, LAI C S, YAN Y L, et al. Histological evidence of chitosan-encapsulated curcumin suppresses heart and kidney damages on streptozotocin-induced type-1 diabetes in mice model [J]. Sci Rep, 2019, 9(1): 15233.

(11) YAO H, SHI H, JIANG C, et al. L-Fucose promotes enteric nervous system regeneration in type 1 diabetic mice by inhibiting SMAD2 signaling pathway in enteric neural precursor cells [J]. Cell Commun Signal, 2023, 21(1): 273.

• Urine volume cannot be specified per mouse (methods 4.4) unless the mice were single-housed or if the different groups were not mixed, both are not ideal study set-ups. Please clarify in the methods section.

Thanks for your constructive suggestion. After successful modeling of T1DM mice, the successful modeling mice were grouped based on method 4.2 as follows Control, T1DM, T1DM + EUG (5 mg/kg/day), T1DM + EUG (10 mg/kg/day), and T1DM + EUG (20 mg/kg/day). To ensure consistency among groups, each group consisted of 5 mice and had equal amounts of diet (100 g), drinking water (250 mL), and environmental conditions for feeding. The urine-soaked area of mice in each group was recorded to quantify the urine volume. The conditions are the same for each group. The description of Method 4.4 has been improved in the revised manuscript.

• OGTT (Figure 1H) of week 2 is missing. This is an important control time point, as it would show the effect of STZ before EUG treatment.

Thanks for your careful review. OGTT (Figure 1H) of week 2 has been added in the revised manuscript.

• In Figure 1J, the control group does not follow the expected ITT trajectory. If possible, add the 120-minute time point to see if the blood glucose levels return to baseline in the control group. The graph shows increased basal glucose levels in the experimental groups, but no differences in insulin tolerance. It also misses the AUC calculations. It is probably not significantly different, which should be noted in the text.

Thanks for your suggestion. T1DM primarily manifests as pancreatic β cell damage and the absolute reduction of insulin secretion, resulting in the disorder of glucose metabolism in vivo. The oral glucose tolerance test (OGTT) is a series of plasma glucose concentrations measured within 2 h after oral gavage of a certain amount of glucose. It is a standard method to evaluate an individual's blood glucose regulation ability and to understand the function of islet β cells. Insulin resistance means reducing the efficiency of insulin to promote glucose uptake and utilization for various reasons, and the body's compensatory secretion of excessive insulin leads to hyperinsulinemia to maintain the stability of blood glucose. The insulin resistance test (ITT) is commonly employed to detect insulin resistance in T2DM. However, it was found that the ITT experiment had little correlation with T1DM. Therefore, the ITT experiment of Figure 1J and related description have been removed from the revised manuscript.

• The staining and FACS data on the effects of STZ+EUG+/- ML385 are not convincing (Figure 6 and Figure 7) and do not seem to align with the bar graphs and the conclusions in the text. It would be good to include immunofluorescent staining for insulin to further validate the effects of STZ+EUG+/- ML385 on insulin expression.

Thanks for your comments.

(1) In the revised manuscript, between the statistical results and the pictures, so we re-conducted the statistics of the immunofluorescence results of NRF2 and HO-1, as follows：

(1) NRF2 immunofluorescence staining:

**Author response image 2. sa3fig2:** Group 1.

**Author response image 3. sa3fig3:** Group 2.

**Author response image 4. sa3fig4:** Group 3.

**Author response image 5. sa3fig5:** Group 4.

**Author response image 6. sa3fig6:** Group 5.

**Author response image 7. sa3fig7:** NRF2 immunofluorescence staining statistics.

(2) HO-1 immunofluorescence staining:

**Author response image 8. sa3fig8:** Group 1.

**Author response image 9. sa3fig9:** Group 2.

**Author response image 10. sa3fig10:** Group 3.

**Author response image 11. sa3fig11:** Group 4.

**Author response image 12. sa3fig12:** Group 5.

**Author response image 13. sa3fig13:** HO-1 immunofluorescence staining statistics.

(2) The meanings represented by each quadrant of cell flow analysis are as follows: Q1 represents a group of necrotic cells, characterized by positive PI staining and negative Anenexin V staining; Q2 represents late apoptotic cells, with both PI and Anenexin V staining negative; Q3 represents early apoptotic cells, with both PI and Anenexin V staining positive; Q4 represents living cells, characterized by positive Anenexin V staining and negative PI staining. In the experiment, the number of apoptotic cells were calculated as the sum of late apoptotic cells in Q2 and early apoptotic cells in Q3. As shown in Figure 9F-G, these results were consistent with those observed in Figure 6G, 6J and Figure 7D-F.

(3) MIN6 cells, as mouse islet β cell line, has the function of secreting insulin. The intervention of STZ was an absolute decrease in the number of islet β cells, so the result of insulin immunofluorescence staining was only a decrease in the number of MIN6 cells in each cell group. In addition, the detection of insulin protein expression level is always through ELISA method to assess the secretion of insulin protein in the cell supernatant. Figure 6E is the ELISA results of insulin protein secretion in the cell supernatant.

• The experimental design for the in vitro experiments was unclear from the text. Consider including a schematic to show when cells were treated with STZ, EUG, and ML385.

Thanks for your suggestion. The experimental design for the in vitro experiments of this study has been added in Figure 6A of the revised manuscript.

• As stated in the Discussion, the use of the insulinoma line Min6 as a model instead of primary pancreatic beta cells is a clear limitation of the study. The mechanistic data would be stronger if validated on a more relevant system (eg. untransformed Islets).

Thanks for your comments. Mouse insulinoma cells (MIN6 cell line) are permanent cell lines isolated from mouse islet β cell tumors, which can reflect the functional changes of islet β cells. As mature islet cells, MIN6 cells have been widely utilized as an in vitro cellular model for diabetes research to investigate the functionality of β cells within pancreatic islets[1, 2, 12]. So in this study, MIN6 cells were used as the cell model in vitro. In our future studies, we will try to conduct our findings using more relevant in vitro cell systems (eg. Islets).

References:

(1) WU M, CHEN W, ZHANG S, et al. Rotenone protects against β-cell apoptosis and attenuates type 1 diabetes mellitus [J]. Apoptosis, 2019, 24(11-12): 879-91.

(2) LUO C, HOU C, YANG D, et al. Urolithin C alleviates pancreatic β-cell dysfunction in type 1 diabetes by activating Nrf2 signaling [J]. Nutr Diabetes, 2023, 13(1): 24.

(12) CHEN H, LOU Y, LIN S, et al. Formononetin, a bioactive isoflavonoid constituent from Astragalus membranaceus (Fisch.) Bunge, ameliorates type 1 diabetes mellitus via activation of Keap1/Nrf2 signaling pathway: An integrated study supported by network pharmacology and experimental validation [J]. J Ethnopharmacol, 2024, 322: 117576.

• The use of small molecule inhibitors such as ML385 can have unspecific effects. Genetic manipulation or the use of siRNAs to inhibit the NRF2 pathway would have been preferable for the in vitro experiments.

Thanks for your constructive suggestion. ML385 is a commonly used and stable inhibitor of the NRF2 and has been used in a variety of disease studies[13-15]. The MIN6 cells utilized in this study were cultured under challenging conditions and exhibited a sluggish growth rate. Owing to the cytotoxicity associated with siRNAs transfection reagents, a significant proportion of MIN6 cells succumbed following transfection. Consequently, small molecule inhibitors ML385 were employed in this investigation. In our future studies, we will try to conduct our findings using siRNAs.

References:

(13) DANG R, WANG M, LI X, et al. Edaravone ameliorates depressive and anxiety-like behaviors via Sirt1/Nrf2/HO-1/Gpx4 pathway [J]. J Neuroinflammation, 2022, 19(1): 41.

(14) WANG Z, YAO M, JIANG L, et al. Dexmedetomidine attenuates myocardial ischemia/reperfusion-induced ferroptosis via AMPK/GSK-3β/Nrf2 axis [J]. Biomed Pharmacother, 2022, 154: 113572.

(15) LI J, DENG S H, LI J, et al. Obacunone alleviates ferroptosis during lipopolysaccharide-induced acute lung injury by upregulating Nrf2-dependent antioxidant responses [J]. Cell Mol Biol Lett, 2022, 27(1): 29.

• The study proposes a mechanism in which EUG-induced disruption of KEAP1 and NRF2 interaction leads to NRF2 translocation to the nucleus and upregulation of proteins required to prevent oxidative stress. In Figure 6H it is unclear whether the nuclear NRF2 increases. Please add quantifications of the immunostainings.

Thanks for your reminder. Figure 6J shows the quantifications of the immunostainings of NRF2 in the revised manuscript.

• Some of the figure legends lack important information. In Figure 5A, 6E for instance, what is the protein expression normalized to?

Thanks for your constructive suggestion. Protein normalization refers to the standardization of proteins from different sources and with different properties, so as to facilitate the comparison of protein content and expression in different samples. In WB experiment, protein expression normalization is one of the essential steps. Western blot of nuclear protein generally cannot be performed using β-Actin as an internal reference. Lamin B was chosen because β-Actin is an intrinsic parameter not found in the nucleus. N-NRF2, as a nuclear protein, requires Lamin B as a reference for protein normalization. The lack important information of WB in Figure have been supplemented in figure legends of the revised manuscript.

• Please acknowledge previous literature on the effects of EUG/clove oil in diabetes models. The meta-analytical review by Carvalho et al. (DOI: 10.1016/j.phrs.2020.105315)should be cited and discussed.

Thanks for your suggestion. It has been cited and discussed in the revised manuscripts.

• Consider revising the text for grammar, language mistakes, and readability. The text is not always precise (e.g. in the explanation of gamma-H2AX in the results), does not explain terminology (e.g. the oxidative stress markers - line 204+205), or simplifies conclusions (e.g. "improved islet function" based on glucose tolerance test", line 129).

Thanks for your comments. The above problem has been solved in the revised manuscripts. In addition, we had send our manuscript to the professional English language editing company to improve our paper, and the editorial certificate had been submitted as a supplement document.

• In the current format, some figures are out of focus. Please make sure to upload a high-quality version for publication.

Thanks for your suggestion. A high quality version figures has been uploaded. Perhaps due to the excessive content of the file after upload, the file is compressed, and the figures is not focused. So, all figures in this study have been uploaded separately for download in the review system.

**Reviewer #2 (Recommendations For The Authors):**
Below are specific points of criticism on the experiments presented.(1a) There is no comparison among eugenol treatments with regards to fasting weight, blood glucose, water intake, food intake, and, crucially, OGTT. All three treatments appear to show very similar effects but has this been statistically assessed? Shown statistical significance of ketonuria between no and high eugenol treatments seems exaggerated.

Thanks for your comments. EUG intervention has a dose-dependent effect on T1DM. According to Figure 1B-I, 20 mg/kg EUG has the best effect. Fasting body weight, blood glucose, water intake, food intake, and OGTT were statistically assessed in Figure 1 of the revised manuscript. In addition, we performed statistical analyse of ketonuria between no and high eugenol treatments again in the revised manuscript. In the revised manuscript, we have also made objective revisions to the expression of eugenol's efficacy.

(b) ITT is not used to detect T1DM (line 126).

Thanks for your suggestion. T1DM primarily manifests as pancreatic β cell damage and the absolute reduction of insulin secretion, resulting in the disorder of glucose metabolism in vivo. The oral glucose tolerance test (OGTT) is a series of plasma glucose concentrations measured within 2 h after oral gavage of a certain amount of glucose. It is a standard method to evaluate an individual's blood glucose regulation ability and to understand the function of islet β cells. Insulin resistance means reducing the efficiency of insulin to promote glucose uptake and utilization for various reasons, and the body's compensatory secretion of excessive insulin leads to hyperinsulinemia to maintain the stability of blood glucose. The insulin resistance test (ITT) is commonly employed to detect insulin resistance in T2DM. However, it was found that the ITT experiment had little correlation with T1DM. Therefore, the ITT experiment and related description have been removed in the revised manuscript.

(2) Here it is hard to reconcile the gradual increase of Ins protein levels in (STZ) and (STZ + increasing eugenol) samples with(a) results in 1 suggesting that the dose of eugenol does not significantly affect the outcome and(b) Ins expression, which is essentially undetectable in both STZ and STZ+EUG mice. A likely explanation is that EUG just postpones beta cell death. I assume that these analyses were done in week 10 but it is not stated.

Thanks for your professional suggestion. Perhaps because the file is compressed, the gray value of WB strip is not obvious, so the expression of INS is not seen clearly. In fact, the intervention of STZ resulted in a significant decrease in INS expression compared with the Control group, which could be alleviated by the treatment of EUG. However, due to the large difference in INS between the STZ group, EUG treatment, and the Control group, the gray values of INS in the STZ group and the STZ + EUG group were not clear. As mentioned in the method 4.12-4.13, our WB and PCR samples were from 10 week mice.

(3) The γH2Ax stainings provided are weak and do not fully correspond to the quantitation - the 5 mg/Kg EUG treatment appears less severe than the 10 mg/Kg. In contrast, changes in the PCD pathway are convincingly demonstrated.

Thanks for your reminder. γH2AX immunohistochemical staining is required to be located in the islets. It measured the number of β cells stained with brown, not the brown area. The ZOOM image of γH2AX staining showed that the EUG improvement effect of 10 mg/kg was better than that of 5 mg/kg. γH2AX, as a marker of DNA damage, exhibits nuclear localization and is absent in the cytoplasmic compartment. Therefore, in Figure 4C-D, we quantified the proportion of cells exhibiting brown staining. In Figure 4C, black arrows were employed to highlight the presence of brown-stained islet β cells.

(4) Is there a reason for looking at mRNA levels of Ho-1 but not KEAP1 or NQO-1 ? What is the expression of Nrf2 itself at the RNA level? Please give in the text what the abbreviations MDA, SOD, CAT GSH-Px stand for. Are these protein levels or activity assays? Units in the y-axis of graphs?

Thanks for your constructive suggestion.The required KEAP1 and NQO-1 primers have been synthesized, and the relevant data have been supplemented in the revised manuscript. The expression of Nrf2 itself at the RNA level is T-NRF2 (Total NRF2). The MDA, SOD, CAT and GSH-Px abbreviations stand for Malondialdehyde, Superoxide dismutase, Catalase, Glutathione peroxidase, and the relevant information, which have been supplemented in the revised manuscript. These are activity assays of serum, and units in the y-axis of graphs have been added in the revised manuscripts.

(5) The Ins levels in the culture medium of STZ + ML treated cells are much lower than the levels in STZ treated cells (6D). This is not consistent with the results of Ins cell content or Ins expression as stated (6B and D).

Thanks for your careful review. The experimental samples in Figure 6C in the revised manuscript represent the proteins extracted from cells of each group, while the experimental samples in Figure 6E represent the supernatant of cells from each group. ML385 is an inhibitor of NRF2, which effectively suppresses the NRF2 signaling pathway and aggravates MIN6 cell damage, resulting in lower INS expression observed in both the STZ+ML385 group depicted in Figures 6C and 6E compared to that in the STZ group. Although the sample sources of the two groups differ and there are slight variations in the trend, it can be observed that the overall trend of the STZ+ML385 group is comparatively lower than that of the STZ group.